# Regulation of Liprin-α phase separation by CASK is disrupted by a mutation in its CaM kinase domain

Debora Tibbe[1], Pia Ferle[1], Christoph Krisp[2], Sheela Nampoothiri[3] ⓘ, Ghayda Mirzaa[4,5,6] ⓘ, Melissa Assaf[7] ⓘ, Sumit Parikh[8], Kerstin Kutsche[1], Hans-Jürgen Kreienkamp[1] ⓘ

**CASK is a unique membrane-associated guanylate kinase (MAGUK) because of its Ca$^{2+}$/calmodulin-dependent kinase (CaMK) domain. We describe four male patients with a severe neurodevelopmental disorder with microcephaly carrying missense variants affecting the CaMK domain. One boy who carried the p.E115K variant and died at an early age showed ponto-cerebellar hypoplasia (PCH) in addition to microcephaly, thus exhibiting the classical MICPCH phenotype observed in individuals with *CASK* loss-of-function variants. All four variants selectively weaken the interaction of CASK with Liprin-α2, a component of the presynaptic active zone. Liprin-α proteins form spherical phase-separated condensates, which we observe here in Liprin-α2 overexpressing HEK293T cells. Large Liprin-α2 clusters were also observed in transfected primary-cultured neurons. Cluster formation of Liprin-α2 is reversed in the presence of CASK; this is associated with altered phosphorylation of Liprin-α2. The p.E115K variant fails to interfere with condensate formation. As the individual carrying this variant had the severe MICPCH disorder, we suggest that regulation of Liprin-α2–mediated phase condensate formation is a new functional feature of CASK which must be maintained to prevent PCH.**

## Introduction

Perturbations in synapse formation, synaptic protein complexes, and synaptic transmission are associated with neurodevelopmental disorders in humans (Grabrucker et al, 2011; Bourgeron, 2015). Both pre- and postsynaptically, large protein complexes are formed which contribute to synaptic architecture and function. One such complex, the presynaptic active zone, consists of a dense network of core constituents ELKS, Liprin-α, RIM, RIM-BP, and Munc13. The active zone complex is held together through multiple interactions via highly conserved domains such as C$_2$, PDZ, and SH3 domains (Sudhof, 2012). Recently, it became clear that active-zone assembly also relies on a process termed liquid–liquid phase separation (LLPS), which involves the recruitment of proteins into condensates mediated by multiple intrinsically disordered regions (IDRs) (Emperador-Melero et al, 2021; Liang et al, 2021; Xie et al, 2021).

The active zone is required for the recruitment of synaptic vesicles to release sites precisely opposite to postsynaptic specializations containing the appropriate neurotransmitter receptors (Sudhof, 2012). This positioning requires transsynaptic adhesion complexes such as the Neurexin/Neuroligin pair of adhesion molecules (Sudhof, 2008). Neurexins are linked to the active zone through the Ca$^{2+}$/calmodulin–dependent serine protein kinase CASK (Hata et al, 1996). CASK binds to Neurexins via its C-terminal PDZ-SH3-GK (PSG) module (Hata et al, 1996; Li et al, 2014; Pan et al, 2021), whereas the N-terminal CaM-dependent kinase domain (CaMK domain) is involved in protein interactions with Liprin-α (Wei et al, 2011; LaConte et al, 2016). Additional interactions with Mint1, also through the CaMK domain and Lin/Veli proteins through the L27.2 domain, establish CASK as a multivalent scaffold protein (Butz et al, 1998; Tabuchi et al, 2002). The CaMK domain exhibits an Mg$^{2+}$-sensitive, atypical kinase activity which may phosphorylate the Neurexin C-terminus (Mukherjee et al, 2008). The functional relevance of this activity is unclear. In addition to its presynaptic role, CASK has been reported to act as a transcriptional regulator during neuronal development (Hsueh et al, 2000) and as a regulator of postsynaptic glutamate receptor trafficking (Jeyifous et al, 2009).

Loss-of-function variants in the X-chromosomal *CASK* gene lead to microcephaly with pontine and cerebellar hypoplasia (MICPCH) and intellectual disability (ID) in females in the heterozygous state and in males in the hemizygous state. Furthermore, several missense variants have been described which are associated with neurodevelopmental disorders of variable severity. These variants mostly affect males and are often inherited from healthy mothers (Najm et al, 2008; Hackett et al, 2010; Pan et al, 2021).

[1]Institute for Human Genetics, University Medical Center Hamburg-Eppendorf, Hamburg, Germany   [2]Institute for Clinical Chemistry and Laboratory Medicine, Mass Spectrometric Proteomics, University Medical Center Hamburg-Eppendorf, Hamburg, Germany   [3]Department of Pediatric Genetics, Amrita Institute of Medical Sciences and Research Centre, Cochin, India   [4]Center for Integrative Brain Research, Seattle Children's Research Institute, Seattle, WA, USA   [5]Department of Pediatrics, University of Washington, Seattle, WA, USA   [6]Brotman Baty Institute for Precision Medicine, Seattle, WA, USA   [7]Banner Children's Specialists Neurology Clinic, Glendale, AZ, USA   [8]Pediatric Neurology, Cleveland Clinic, Cleveland, OH, USA

Correspondence: Kreienkamp@uke.de

So far, the pathogenic mechanisms of *CASK* mutations remain unclear; in particular, it is unknown why some variants cause ID, whereas others are associated also with pontocerebellar hypoplasia (PCH). As CASK fulfils multiple functions at the pre- and postsynapse and in the nucleus, we do not know which of these apparently separate functions contributes most strongly to the patient's phenotype. We have begun to address this by analysing a larger number of missense variants with respect to interactions with a panel of known CASK-associated proteins. Our initial data indicated that the presynaptic role of CASK was affected in most cases as most variants interfered with Neurexin binding (Pan et al, 2021). Here, we identify four male patients carrying missense variants in the CaMK domain of CASK. All four variants selectively interfere with Liprin-α2 binding, strongly supporting a disturbed presynaptic function of CASK as a major pathogenic mechanism. Importantly, we also observe that in human cells and in neurons, Liprin-α2 undergoes formation of spherical condensates. In the presence of CASK-WT, but not a CASK variant associated with a severe phenotype, condensate formation is reduced. Our data uncover a new aspect of the molecular function of CASK which may be relevant for proper formation of the presynaptic active zone.

# Results

### Missense variants altering the CaMK domain of CASK

Four novel hemizygous missense variants (p.E115K, p.R255C, p.R264K, and p.N299S) in *CASK* were identified in male patients, leading to substitutions in the CaMK domain (Fig 1A). All four patients were severely affected by microcephaly, severe developmental delay, intellectual disability, and seizures (Table 1). Patient 1 (p.E115K) stands out as he additionally showed PCH and thus had the MICPCH disorder, usually associated with *CASK* loss-of-function mutations. This patient died at a young age. Previously, only three further missense variants in the CaMK domain have been reported, namely p.G178R, p.L209P, and p.Y268H (Hackett et al, 2010; LaConte et al, 2019; González-Roca et al, 2020). As it is unclear how N-terminal variants affect protein function, we analysed the functional relevance of the new variants identified here.

To assess the potential of the CASK variants to disrupt specific functions of CASK, we looked at 3D crystal structures of the CASK CaMK domain in complexes with Mint1 and Liprin-α2, two prominent presynaptic partners of CASK. Both Mint1 and Liprin-α2 use the insertion of a tryptophan side chain which is part of a conserved Ile/Val-Trp-Val sequence (Trp981 in Liprin-α2) into a deep hydrophobic pocket in CASK (Wei et al, 2011; Wu et al, 2020). This pocket is formed between the αD and αE helices of the kinase domain (Fig 1B). As the αD helix is intimately connected to the αR1 helix in CASK, it is conceivable that the N299S variant in αR1 might alter the hydrophobic pocket. The E115K variant in αE is likely to affect the position of αE, thereby changing the size of the pocket. Liprin-α2 requires a second interface for high-affinity binding, which is formed by its SAM2 domain. For CASK, this involves part of the CaMK surface containing residues R255, R264, and Y268. The substitutions

R255C and R264K (as well as Y268H, analysed by Wei et al [2011]) therefore are likely to affect the second interface between CASK and the SAM2 domain of Liprin-α2.

### Binding to Mint1 and Neurexin is not affected, whereas binding to Veli and ATP is slightly altered by missense variants affecting the CASK CaMK domain

CASK forms an evolutionarily conserved trimeric complex with Mint1 and Veli proteins (Butz et al, 1998). The interaction between CASK and Mint1 is mediated by the CaMK domain of CASK (Wu et al, 2020; Zhang et al, 2020). We tested whether the variants altered binding of CASK to Mint1 or Veli. HEK293T cells were cotransfected with plasmids coding for mRFP-tagged CASK variants (or mRFP alone as the negative control) and GFP-tagged Mint1. Veli proteins are highly expressed endogenously in HEK293T cells and were visualized in two bands at about 26 and 30 kD, corresponding to Veli1 at 30 kD and Veli2/3 at 26 kD. Upon cell lysis and immunoprecipitation of mRFP-containing proteins, we found that all CASK mutant variants interact consistently well with Mint1 and Veli proteins. Veli2/3 were more prominent in the IP sample compared with Veli1 (Fig 2A). The E115K variant coprecipitated slightly more Veli proteins than CASK-WT and the remaining mutants (Fig 2A–C).

The C-terminal PDZ ligand of Neurexins binds to the PDZ domain of CASK (Hata et al, 1996). Structural work has shown that the full PSG superdomain is necessary for high-affinity interaction (Li et al, 2014) and oligomerization (Pan et al, 2021). Importantly, the C-terminus of Neurexin is the only known in vivo substrate for the CASK CaM kinase activity (Mukherjee et al, 2008; Mukherjee et al, 2010), suggesting the possibility that alterations in the CaMK domain might affect the interaction between both proteins. To test this, HEK293T cells were transiently transfected with plasmids coding for mRFP-CASK and HA-Neurexin-1β. As before, mRFP-tagged CASK variants (or mRFP as control) were immunoprecipitated from cell lysates. Amounts of CASK and coprecipitated Neurexin were analysed by Western blot and quantified (Fig 2D and E). With all CASK variants, Neurexin-1β was coimmunoprecipitated efficiently, and no differences with respect to the CASK-Neurexin interaction were detected.

For measurement of ATP binding to the active site of the kinase domain, we employed the fluorescent ATP analogue TNP-ATP. WT and mutant CASK kinase domains were expressed as His$_6$-tag-SUMO fusion proteins in bacteria and purified. Binding of TNP-ATP to both WT and mutant CASK kinase domains could be verified by a shift of the fluorescence spectrum of TNP-ATP to lower wavelengths and an increase in fluorescence intensity (Fig 3A). TNP-ATP binding was reduced in the presence of $Mg^{2+}$, in keeping with the designation of CASK as an atypical, $Mg^{2+}$-sensitive instead of $Mg^{2+}$-dependent kinase (Mukherjee et al, 2008). ATP binding to all CASK variants proved to be $Mg^{2+}$-sensitive. By calculating the ratio of fluorescence in the absence and in the presence of $Mg^{2+}$ as the "$Mg^{2+}$ sensitivity," we determined that the R264K and N299S variants of CASK bound more TNP-ATP in the absence of $Mg^{2+}$ and were significantly more sensitive to $Mg^{2+}$ than the WT protein and the mutant E115K. The R255C variant showed lower binding to TNP-ATP in the absence and presence of $Mg^{2+}$, resulting in an unchanged "$Mg^{2+}$ sensitivity" compared with CASK-WT (Fig 3B and C).

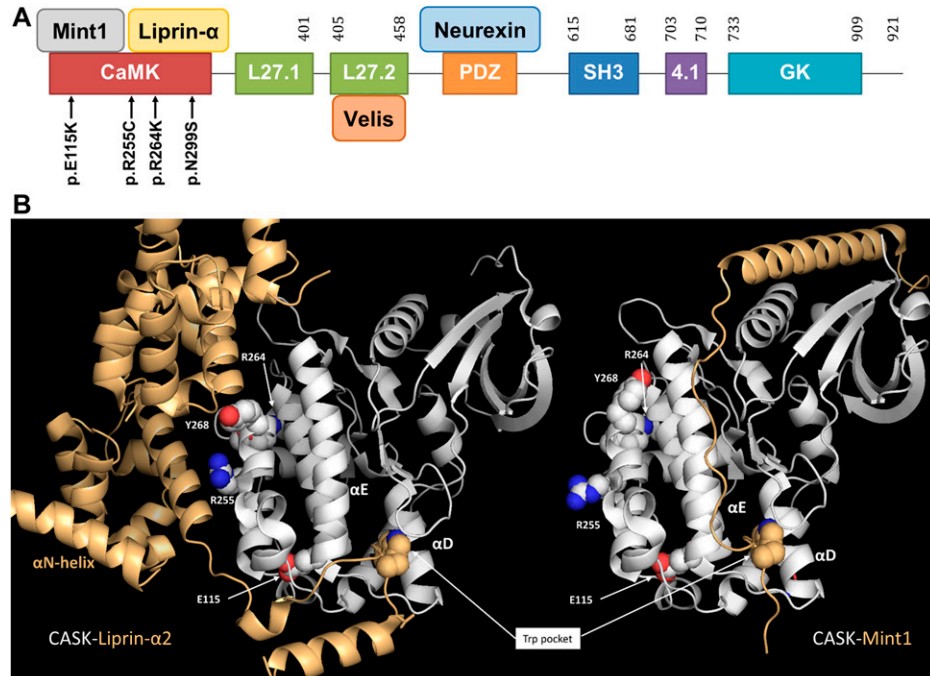

**Figure 1. Identification of missense variants affecting the CaMK domain of CASK.**
**(A)** Protein domains of CASK, selected interaction partners, and missense variants in the *CASK* gene identified here. Numbering refers to the database entry NP_003679.2. **(B)** Comparison of CASK-Liprin-α2 and CASK-Mint1 complexes. The positions of residues analysed here is indicated, with the exception of N299 in the αR1-helix which is occluded by the αD-helix. Note that while the hydrophobic tryptophan pocket is occupied in an identical manner in both complexes, both complexes differ in their secondary binding site, with a cluster of mutants (R255C, R264K, and Y268H analysed by Wei et al [2011]) affecting only the secondary site for Liprin-α2. For Liprin-α2, the position of the αN-helix is also indicated which is pointing away from the SAM domains upon complex formation with CASK. Structures were derived from database entries 3tac in case of Liprin-α2 (Wei et al, 2011) and 6kmh in case of Mint1 (Zhang et al, 2020) and visualized with Pymol.

## All CASK variants exhibit reduced binding to Liprin-α2

To analyse whether binding to the active zone component Liprin-α2 (Olsen et al, 2005; LaConte et al, 2016) is affected, HEK293T cells were transfected with plasmids coding for either a RFP-tagged CASK variant or the empty vector control together with a construct coding for HA-tagged Liprin-α2. Upon immunoprecipitation of mRFP-tagged proteins, we found that all four tested CaMK mutants showed a significant reduction in interaction with Liprin-α2 compared with CASK-WT (Fig 4A and B). We repeated this assay in a different format, using an immobilized SUMO fusion protein of the three C-terminal SAM domains of Liprin-α2 (tSAM), which contain the CASK binding loop (Wei et al, 2011), in a pulldown assay from HEK293T cells expressing the different mRFP-tagged CASK variants. Here, we similarly observed differences between CASK-WT and the four mutants as significantly reduced binding was detected for E115K, R255C, and R264K. The reduction in binding for the N299S variant was not significant after three repeats (Fig 4C and D).

One might ask why binding of CASK to Liprin is reduced by the variants, whereas binding to Mint1 is not. Mint1 also uses the hydrophobic pocket of CASK for insertion of its Val-Trp-Val sequence; however, Mint1 uses a second binding interface distinct from that of Liprin-α2. This involves a stretch of α-helix which makes an extensive contact to the N-lobe of the CASK CaMK domain (Fig 1B) (LaConte et al, 2016; Wu et al, 2020), allowing for a much higher affinity of the Mint1-CASK interaction (Wu et al, 2020). We think that Mint1 is not affected by the variants because (a) R255 and R264 are in the second interface for Liprin-α2 but not for Mint1 (Fig 1B); and (b) because the second interface for Mint1 provides more strength to the interaction. Thus, disruption of the hydrophobic pocket for the Trp side chain can be partially compensated by the second interface of Mint1, but not of Liprin-α2.

We focused further on the interaction between CASK and Liprin-α2. In Fig 4A, and more pronounced in Fig 5A, we noted that the Liprin-α2 band in Western blot analysis was smeared and was shifted to higher molecular weights when coexpressed with mRFP from the empty vector but not when coexpressed with CASK-WT. Upward smearing of Liprin-α2 was also observed when CASK-E115K was coexpressed (see also quantitative analysis of Western blot bands in Fig S1). R264K and N299S showed a somewhat intermediary effect (Fig 5A). As we suspected a phosphorylation event, lysates of cells expressing mRFP-CASK or mRFP alone together with Liprin-α2 were treated with FastAP phosphatase. This treatment eliminated the shift to higher molecular weight (Fig 5B). Coexpression of CASK-WT with Liprin-α2 led to higher electrophoretic mobility of Liprin-α2, which was similar to that of the phosphatase treated samples in the absence of CASK. No further increase in mobility was observed when the CASK + Liprin-α2 samples were treated with phosphatase (Fig 5B). As a conclusion, Liprin-α2 is phosphorylated in 293T cells in the absence of CASK; coexpression of CASK-WT but not CASK-E115K interfered with this phosphorylation event in Liprin-α2.

We determined phosphorylation sites by mass spectroscopic analysis of immunoprecipitated Liprin-α2, isolated from cells coexpressing mRFP control vector, or coexpressing mRFP-tagged CASK. Numerous phosphorylated sites were detected in the N-terminal coiled-coil regions (i.e., S73, S87, S257, S260, S263) and the intervening IDR (S552 and S673). Remarkably, phosphorylation of S87 in coiled-coil region 1 was decreased when CASK was coexpressed in three independent repeats (Fig 5C and Table 2).

Liprin-α proteins from various species form condensates in cells through a process termed LLPS (McDonald et al, 2020; Emperador-Melero et al, 2021; Liang et al, 2021; Xie et al, 2021). During LLPS, proteins form dynamic, non-membrane surrounded compartments through demixing from the diffuse state out of the cytosol (Bracha

**Table 1. Clinical features in four male patients with a *CASK* missense variant.**

| Patient # | 1 | 2 | 3 | 4 |
|---|---|---|---|---|
| Variant (NM_003688.3) | c.343G>A p.(Glu115Lys) | c.763C>T p.(Arg255Cys) | c.791G>A p.(Arg264Lys) | c.896A>G p.(Asn299Ser) |
| Inheritance | de novo | Maternally inherited (somatic mosaicism) | Unknown (adopted) | Maternally inherited |
| Sex | Male | Male | Male | Male |
| Pregnancy and birth | | | | |
| Pregnancy | Unremarkable | Class III obesity; hypothyroidism – levothyroxine; anxiety/depression – Zoloft Iron; prenatal multivitamin; scheduled repeat caesarean section | Delivered at 34 wk of gestation, twin pregnancy; twin sister is alive and well without any medical or neurological issues; previous medical history is limited because of the social circumstances as the child has been adopted and there is limited contact with biological parents | Uncomplicated pregnancy; birth via spontaneous vaginal delivery |
| Birth at | Full term | 39 wk, NICU for 5 wk | 34 wk | 37 wk |
| Birth weight (centile, z-score[a]) | 1,750 g (<3rd centile, −4.25 z) | 2,480 g (1st centile, −2.32 z) | ND | 3.27 kg (65th centile, 0.38 z) |
| Birth length (centile, z-score[a]) | ND | 46.5 cm (1st centile, −2.57 z) | ND | 49.5 cm (31st centile, −0.29 z) |
| OFC birth (centile, z-score[a]) | ND | 32 cm (1st centile, −2.54 z) | ND | ND |
| Last examination | | | | |
| Age | 39 mo | 19 mo | 14 yr | 3 yr 4 mo |
| Weight (centile, z-score[b]) | 9 kg (<3rd centile, −4.21 z) | 9.5 kg (2nd centile, −1.98 z) | 37.2 kg (2nd centile; −1.98 z) | 14.1 kg (26th centile, −0.65 z) |
| Height (centile, z-score[b]) | ND | 79.5 cm (7th centile, −1.49 z) | 134 cm (<1st centile; −3.81 z) | 94.5 cm (11th centile, −1.25 z) |
| OFC (centile, z-score[b]) | 40 cm (<3rd centile, −8.64 z) | 39 cm (<1st centile, −8.11 z) | 49.7 cm (<1st centile; −3.94 z) | 46.2 cm (<1st centile, −3.88 z)[b] |
| Development | | | | |
| DD/ID | Severe global | Severe delays (sometimes rolling back to belly, showing an interest in toys, making some eye contact, moving arms and legs, able to sit with moderate to significant support) | Severe global | Severe global |
| Motor development | No head control | Severe delays (does not sit independently) | Non-ambulatory | Sits without support as of age 2; does not reach or grasp, no purposeful hand use |
| Speech impairment | Non-verbal | Severe delays (nonverbal, no social smiles or interactive speech) | Non-verbal | Vocalizes only |
| Neurological features | | | | |
| Muscular hypotonia and/or hypertonia | Hypotonia | Severe hypotonia (sits with moderate support in tripod position, head is facing down, trying to roll, does not push up from belly) | Increased tone in the upper and lower extremities, specifically with more involvement of the lower extremities; Deep tendon reflexes were 3+ in the lower extremities and 2+ in the upper extremities; kyphoscoliosis | Hypotonia, severe, diffuse, axial and appendicular |

**Table 1. Continued**

| Patient # | 1 | 2 | 3 | 4 |
|---|---|---|---|---|
| Seizures | Yes | No | Yes | Yes |
| Seizure onset | 9 mo | ND | ND | 18 mo |
| Seizure type | Stiffening of upper and lower limbs with occasional head drops | ND | Severe complex epilepsy characterized by tonic-clonic seizures, absence, tonic and myoclonic | Generalized tonic seizures 2–3× a week; history of infantile spasms |
| EEG | Bilateral parieto-temporal epileptiform abnormalities | Normal at 18 mo of age | Abnormal: Frequent interictal generalized epileptiform transients, often occurring in runs, background activity: diffusely slow and poorly organized | Slow spike and wave complexes, multiregional sharps and continuous generalized slow |
| Response to treatment | No improvement in development after initiation of therapy for seizures | ND | Intractable | Infantile spasms resolved on Vigabatrin; currently 2–3 seizures a week on valproic acid |
| MRI or CT scan | Pontocerebellar hypoplasia with mild pontine hypoplasia, moderate hypoplasia of inferior vermis and severe cerebellar hypoplasia; small cerebellar peduncles; mild cerebral atrophy; hypoplasia of optic nerve, optic tracts, and optic chiasma | Brain MRI at age 1 mo: normal; Brain MRI at age 18 mo: somewhat delayed myelination, volume loss with resultant mild prominence of the ventricular system, suspected arachnoid cyst in the posterior fossa | Microcephaly with cerebellar vermis hypoplasia that spared the cerebellar hemispheres; the cortical gyral pattern appeared normal; cerebellar basal ganglia and thalami were normal, and the cortical gyral pattern otherwise appeared normal overall and did not show any distinctive cortical brain malformations | Arachnoid cyst of the left cerebellopontine angle |
| Other findings | | | | |
| Hearing | Auditory evoked response study showed poorly formed wave I from right ear and poorly formed wave I, III and V from left ear at 85 dB | No known abnormalities | ND | Unable to test; no auditory brainstem response completed |
| Eye findings | Fundus evaluation showed bilateral optic disc pallor; visual evoked potential showed poorly formed waveforms from both eyes | Does not track, bilateral nystagmus, bilateral cortical blindness | Optic nerve hypoplasia, cortical visual impairment | Cortical vision impairment |
| Feeding | On oral feeds | G tube-dependent | History of feeding difficulties, including dysphagia, and episodic vomiting; these issues have been stable over time | Pureed foods by mouth |
| Craniofacial dysmorphism | Brachycephaly, large ears, micrognathia, strabismus, short nose, finger joint hypermobility | No | Deep-set eyes with blue irises bilaterally, relatively large mouth with increased space between the teeth, ears appeared mildly laterally prominent | Myopathic and dull facies |
| Additional findings | Dysphagia for liquids, deceased at age 4 yr 9 mo | Murmur of ventricular septal defect, obstructive sleep apnoea requiring oxygen at night | Short stature, sleep issues | Left moderate hydronephrosis because of ureteropelvic junction obstruction |

[a]Centiles and z-scores of birth parameters were calculated based on data of Fenton and Kim (2013).
[b]Centiles and z-scores were calculated according to Kromeyer-Hauschild et al (2001) Monatsschr Kinderheilkd 149: 807. 10.1007/s001120170107. DD, developmental delay; ID, intellectual disability; mo, months; ND, no data; NICU, neonatal intensive care unit; OFC, occipitofrontal head circumference; yr, year(s).

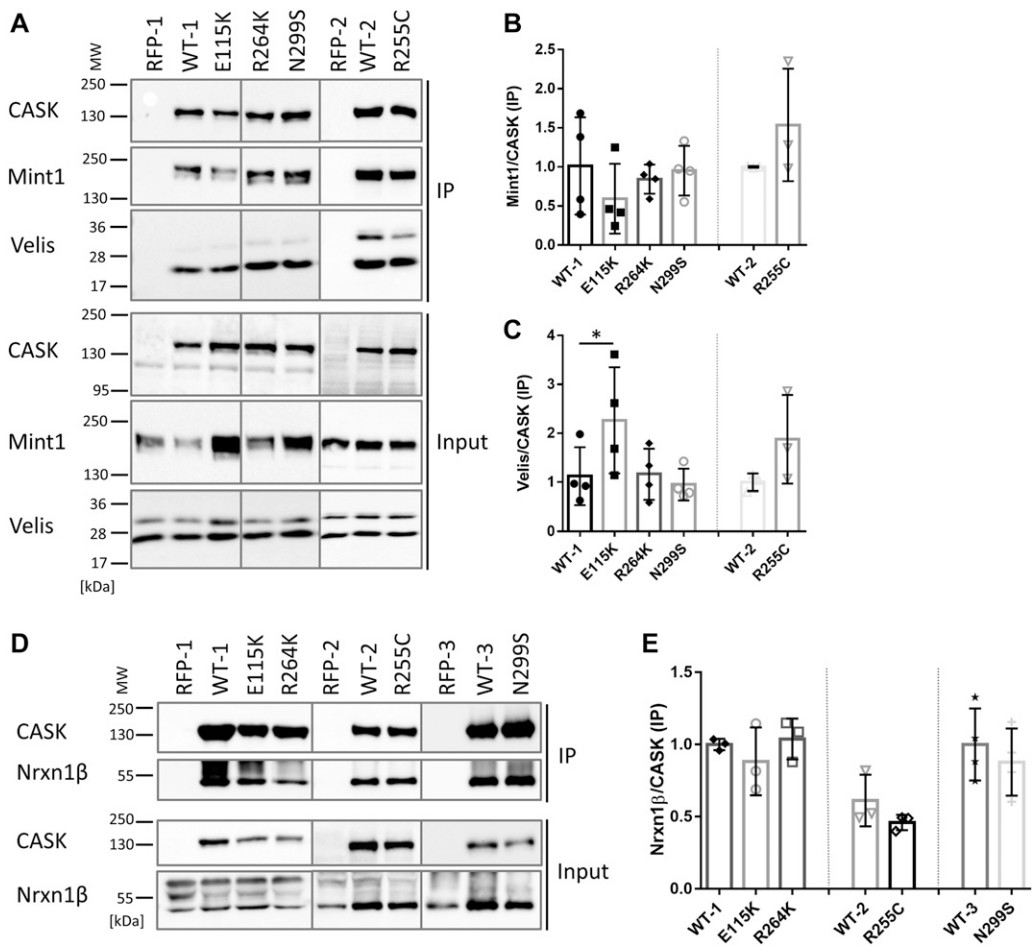

**Figure 2. Substitutions in the CaMK domain of CASK do not alter interactions with Mint1 and Neurexin, whereas binding to Veli is slightly increased for the E115K variant.**

**(A)** mRFP-tagged CASK-WT and mutants or mRFP alone were coexpressed with Mint1 in HEK293T cells. mRFP-tagged proteins were immunoprecipitated from cell lysates, and inputs (IN) and precipitates (IP) were analysed by Western blotting using antibodies against GFP- and mRFP-tags, and anti-Veli. **(B, C)** Quantitative analysis of results shown in (A). Coprecipitation efficiency was determined as the ratio of Mint1 (B) or Veli (C) IP signal, divided by the CASK IP signal. **(D, E)** mRFP-tagged CASK variants were coexpressed with HA-tagged Neurexin, and mRFP-tagged proteins were immunoprecipitated as in (A). Quantitation in (E), performed as in (B) and (C), shows that Neurexin-1β binding was not affected by CASK variants. The mean ± *SD* is shown with each data point representing an independent transfection experiment. Significance was determined by one-way ANOVA with post hoc Dunnett's multiple comparisons test or two-tailed *t* test; *$P \leq 0.05$; n = 3–4.

et al, 2019). The homologue of Liprin-α in *Caenorhabditis elegans* (SYD-2) exhibits LLPS in early stages of synapse development (McDonald et al, 2020). Upon microscopic analysis of Liprin-α2 expressing 293T cells, we observed large spherical clusters of Liprin-α2 which appeared to be phase-separated condensates, such as those observed by Emperador-Melero et al (2021) (Fig 6A and B). Coexpression of Liprin-α2 with CASK-WT resulted in a diffuse cytosolic localization for both proteins in a large proportion of analysed cells. This CASK-dependent change of intracellular localization from bigger condensates to a cytosolic diffuse localization was also observed upon coexpression with CASK variants R255C, R264K, and N299S (Fig 6A and C). In a striking contrast, the localization of Liprin-α2 resembled that observed in the absence of overexpressed CASK when the CASK-E115K variant was coexpressed. CASK-E115K colocalized with Liprin-α2 in the large condensates that were formed (Fig 6A and B). Upon a more detailed analysis of cluster number and cluster size, we observed that CASK variants R264K and N299S displayed an

intermediate phenotype in this assay with numerous clusters of small size (Fig S2). In summary, CASK was able to negatively regulate condensate formation of Liprin-α2, and CASK-E115K failed to do so.

We performed several further experiments to investigate this phenomenon. Firstly, we performed a time resolved series of experiments. Here, cells were first transfected with the Liprin-α2 construct to allow for formation of spherical droplet-like clusters. On the next day, cells were transfected again with *CASK* expression vectors. Here, we observed that CASK-WT was indeed able to "dissolve" preformed condensates of Liprin-α2 (Fig 7).

Liprin-α condensates may have more or less liquid properties, as observed for Liprin-α3 and for Liprin-α2, respectively (Emperador-Melero et al, 2021). To distinguish between these possibilities, we performed FRAP experiments for Liprin-α2 clusters under all conditions analysed here; this includes the rare clusters seen upon coexpression with CASK-WT and variants R255C, R264K, and N299S. In all cases, we observed no recovery of fluorescence (Fig S3), in

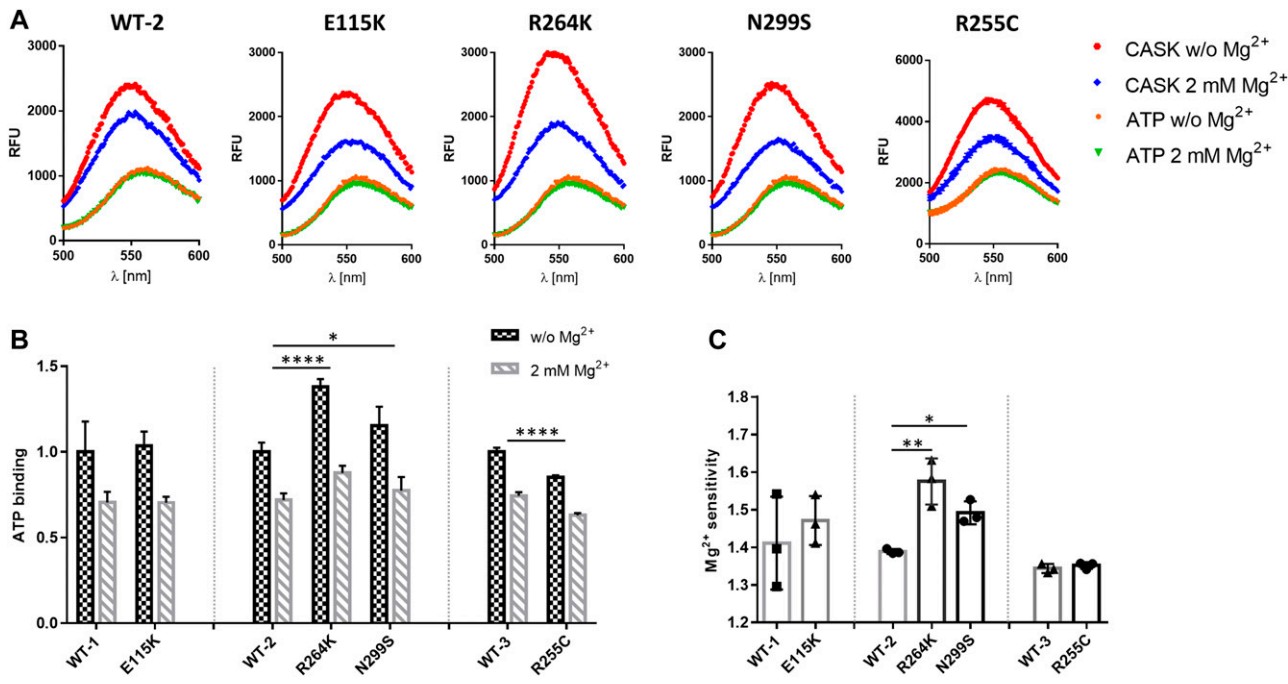

**Figure 3. Mg²⁺ sensitivity of ATP binding is slightly altered by R264K and N299S variants of CASK.**
**(A)** SUMO-tagged fusion proteins of the CaMK domain of CASK were purified and incubated with the fluorescent ATP analogue TNP-ATP in the absence or presence of 2 mM Mg²⁺, and fluorescence emission spectra were recorded from 500 to 600 nm. **(B, C)** Quantification of the maxima of fluorescence signal in the absence and presence of Mg²⁺ (B) and of the ratio between the two values, defined as the "Mg²⁺ sensitivity" (C). Significance was determined by (B) two-way ANOVA with Sidak's multiple comparison test or (C) one-way ANOVA with Dunnett's multiple comparisons test or two-tailed $t$ test; *$P \leq 0.05$, **$P \leq 0.01$; ****$P \leq 0.0001$; n = 3. Mean ± *SD* is shown. In (C), each data point represents independent fusion protein purification.

agreement with the rather gel-like properties of Liprin-α2 clusters described by Emperador-Melero et al (2021).

As we observed a correlation between Liprin-α2 phosphorylation at Ser87 and the formation of condensates, we generated phospho-mimic (S87E) and phospho-dead (S87A) mutants of Liprin-α2; upon expression in 293T cells, we did not observe any alterations in the behaviour of Liprin-α2 with respect to the formation of condensates (Fig S4). These data indicate that phosphorylation at this position is not causative for formation or dissolution of condensates.

In the next set of experiments, we asked how CASK and Liprin-α2 affected their mutual localization in primary-cultured transfected hippocampal neurons. Here, Liprin-α2 expressed alone was found in clusters throughout the cell bodies, dendrites, and axons of transfected neurons. Upon coexpression of CASK-WT, this situation changed as both Liprin-α2 and CASK were localized in a diffuse pattern throughout the cell. Again, the CASK-E115K variant failed to alter the distribution of Liprin-α2, as before in 293T cells (Fig 8). Contrary to the situation in 293T cells, also the R255C and N299S variants had a significant effect on the distribution of Liprin-α2 in transfected neurons (Fig S5). Closer inspection of axons showed that CASK-WT and Liprin-α2 frequently colocalized at vGlut1-positive, presumably presynaptic terminals. In contrast, the large clusters observed in cells coexpressing CASK-E115K and Liprin-α2 were found along the axon but were not colocalized with vGlut1, indicating that these are not functional synaptic sites (Fig 9).

Condensate formation of Liprin-α proteins has been documented in several studies in neurons and in non-neuronal cells;

nevertheless, we suspected that this may be a result of high levels of expression of recombinant Liprin-α2. We therefore varied the amount of expression vectors used for neuronal transfection. Here, we observed that a tenfold reduction of the Liprin-α2 plasmid reduced the proportion of transfected neurons containing large condensates, whereas a twofold reduction had no effect. However, even at the lowest concentration of DNA, condensate formation occurred in about 50% of cells. Similarly, reduction of the amount of *CASK* DNA interfered only partially with its ability to reverse condensate formation (Fig S6).

MAGUKs like CASK or PSD-95 are known to oligomerize through their C-terminal PSG tandem domains (McGee et al, 2001; Rademacher et al, 2019; Pan et al, 2021). We analysed the relation between CASK oligomerization and condensation of Liprin-α2, by using a split-YFP fluorescent complementation assay. Two *CASK* cDNA variants were expressed, carrying either the N-terminal or C-terminal half of YFP. We have shown before that upon assembly of CASK oligomers, this leads to complementation of YFP fluorescence which is detectable in FACS-based assay format (Pan et al, 2021). We observed here low levels of YFP fluorescence when the *CASK* WT or variant constructs were expressed alone in 293T cells. Coexpression of Liprin-α2 led to a strong increase in fluorescence for CASK-WT and the R255C, R264K, and N299S variants but not for the E115K variant (Fig 10). These data suggest the existence of two different states for Liprin-α2: the Liprin-CASK complex which is characterized by diffusely localized oligomers; and Liprin-α2 alone which forms phase-separated condensates. The E115K variant fails

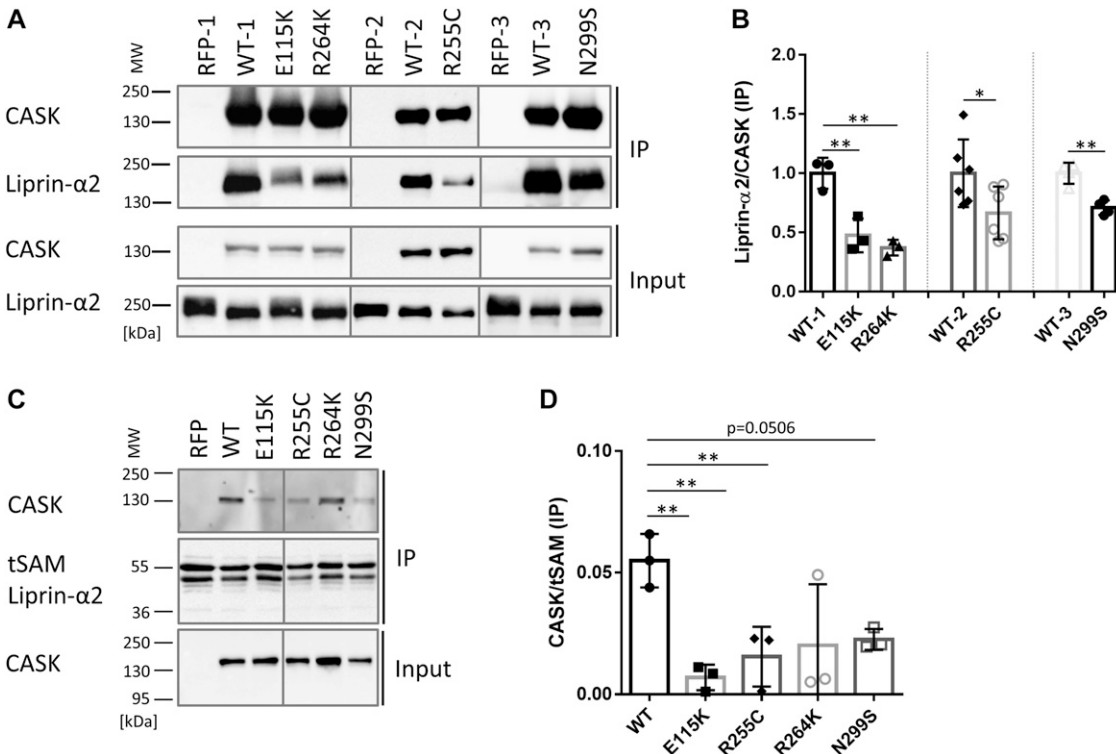

**Figure 4. All four CASK variants interfere with binding to Liprin-α2.**
**(A)** HA-tagged Liprin-α2 was coexpressed with mRFP or mRFP-tagged CASK variants. mRFP-containing proteins were immunoprecipitated from cell lysates, and input and precipitate samples were analysed by Western blotting using epitope-specific antibodies. **(B)** Quantification of the data shown in (A) as mean ± SD. Precipitation efficiency is in each case quantified as the ratio of precipitated HA-Liprin-α2 signal divided by precipitated mRFP-CASK signal. **(C)** A His₆-Sumo fusion protein of the region encompassing the C-terminal three SAM domains of Liprin-α2 (tSAM) was isolated from bacteria using Ni-NTA agarose and left on agarose beads for use in a pulldown assay. Beads were incubated with lysates from cells expressing mRFP alone or mRFP-tagged CASK variants. After washing, input and precipitate samples were analysed by Western blotting using the antibodies indicated. **(D)** Quantification of the data in (C) shown as mean ± SD. Interaction was quantified as the ratio of CASK signal in precipitates, divided by the signal of the tSAM domains. Statistics in (B) and (D) performed with the two-tailed *t* test or one-way ANOVA followed by Dunnett's multiple comparison test, respectively; *$P ≤ 0.05$, **$P ≤ 0.01$; n = 3–6.

to dissolve these condensates into the more diffusely localized CASK-Liprin oligomers.

We sought additional proof that the dissolution of Liprin-α2 condensates by CASK occurs because of the direct interaction of CASK with Liprin. We made use of a mutant with an amino acid exchange in the linker region between Liprin-α2 SAM domains 1 and 2, W981A. This alteration specifically disrupts binding to CASK, without disrupting other functions of the Liprin-α2 SAM domains (Wei et al, 2011). By coexpression of WT and mutant Liprin-α2 with CASK, followed by coimmunoprecipitation, we confirmed that this substitution indeed eliminated the CASK-Liprin interaction (Fig 11A and B). In 293T cells, we observed that W981A mutant Liprin-α2 formed clusters very similar to the WT protein and that coexpressed CASK-WT was unable to interfere with this condensate formation. Despite the complete loss of interaction seen in the biochemical experiment, CASK was recruited to these clusters where it extensively colocalized with Liprin-W891A (Fig 11C and D). We reproduced these results in transfected neurons, where the Liprin-α2 mutant formed clusters throughout the cell. Coexpression of CASK did not alter this localization, and CASK was again found in the same clusters as Liprin-α2 (Fig 11E and F). Thus, we conclude that, irrespective of the cell type, Liprin-α2 forms

spherical clusters, and a tight interaction with CASK is required to regulate cluster formation.

## Discussion

We identified four male patients carrying *CASK* missense variants affecting the CaMK domain. All patients had a severe neurodevelopmental disorder, characterized by microcephaly, intellectual disability, and seizures. Patient 1, carrying the E115K variant, in addition showed pontine and cerebellar hypoplasia, a hallmark of the MICPCH phenotype described for *CASK* loss-of-function variants (Najm et al, 2008; Moog et al, 2015). This patient died at a young age. By performing a thorough functional analysis of all four variants, our goal was to determine which of the various functional aspects of the CASK CaMK domain is responsible for the patients' phenotype. In addition, we wanted to find out what was special about the E115K variant as it causes the additional severe PCH phenotype.

None of the variants appeared to affect folding or stability of overexpressed CASK protein. All four variants did not significantly affect binding to Mint1, to Veli proteins, and to Neurexin.

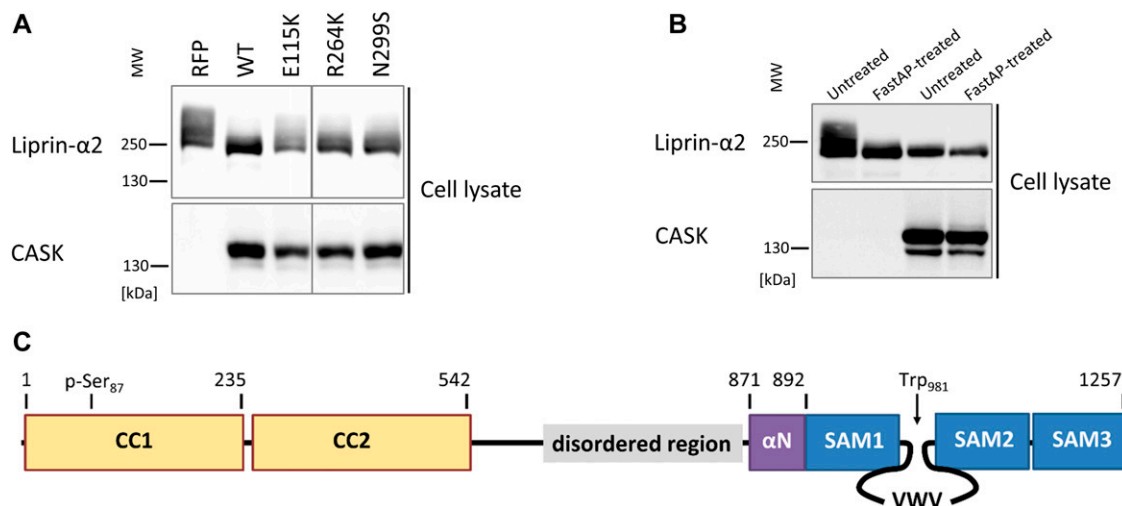

**Figure 5. Interaction with CASK alters the phosphorylation status of Liprin-α2.**
**(A)** Lysates from cells coexpressing HA-tagged Liprin-α2 with mRFP or mRFP-tagged variants of CASK were analysed by SDS–PAGE on an 8% gel, followed by Western blotting. Note the upward smear of the Liprin-α2 specific band, which is abolished by coexpression with CASK-WT but not by CASK mutants. **(B)** Lysates from cells coexpressing HA-tagged Liprin-α2 with mRFP or mRFP-tagged CASK-WT were treated with or without the FastAP alkaline phosphatase, followed by analysis by SDS–PAGE on an 8% gel and Western blotting. **(C)** Domain structure of Liprin-α2; the positions of Ser87 which is most strongly phosphorylated in the absence of CASK, and of Trp981, which constitutes a major binding interface for CASK, are indicated. CC, coiled coil.

Furthermore, none of the variants interfered with folding of the isolated CaMK domain prepared from bacteria. This allowed us to measure $Mg^{2+}$-sensitive ATP binding. $Mg^{2+}$ sensitivity instead of $Mg^{2+}$ dependence classifies CASK as an atypical kinase (Mukherjee et al, 2008). Efficient binding was detected for all four mutants, similar to the wildtype, though variants R264K and N299S slightly altered the $Mg^{2+}$ sensitivity of ATP binding. N299 is located in the kinase regulatory segment in the αR1 helix. A movement of this helix with respect to the rest of the domain, induced by the N299S variant, would change the geometry of the kinase active site, leading to altered binding parameters.

All four variants shared a strongly reduced ability to interact with Liprin-α2, suggesting that a weakened CASK-Liprin connection is responsible for the phenotype of all four patients. Together with our previous findings, showing that loss of Neurexin binding or Neurexin-induced oligomerization is a frequent result of pathogenic *CASK* missense variants (Pan et al, 2021); this points strongly to a presynaptic origin of *CASK*-related neurodevelopmental disorders. But what makes the E115K variant so devastating?

Liprin-α proteins have been shown to be early organizers of presynaptic development by recruiting ELKS, RIM, and CASK (Dai et al, 2006; Spangler et al, 2013). The ability of Liprin-α proteins to undergo LLPS has now been documented in several studies; LLPS is driven by multimerization of its N-terminal coiled-coil motifs, which leads to a high local concentration of Liprin-α and associated ELKS molecules (Liang et al, 2021). Furthermore, a central IDR (see Fig 5C) in Liprin-α proteins from different species contributes to LLPS

**Table 2. CASK coexpression alters phosphorylation of Liprin-α2.**

| Position | Tryptic peptide | Modification | Position in Liprin-α | Exp1 | Exp2 | Exp3 |
|---|---|---|---|---|---|---|
| 63–75 | [R].LQDVIYDRD**S**LQR.[Q] | 1xPhospho [S10] | 1xPhospho [S72] | Up | Down | Up |
| 76–96 | [R].QLNSALPQDIE**S**LTGGLAGSK.[G] | 1xPhospho [S12] | 1xPhospho [S87] | n.d. | Down | n.d. |
| 76–96 | [R].QLNSALPQDIE**S**LTGGLAGSK.[G] | 1xPhospho [S12] 1xGln->pyro-Glu [N-Term] | 1xPhospho [S87] | Down | Down | Down |
| 255–279 | [K].RLSNGSIDSTDETSQIVELQELLEK.[Q] | 1xPhospho [T/S] | 1xPhospho [T/S] | n.d. | n.d. | n.c. |
| 255–279 | [K].RL**S**NG**S**ID**S**TDETSQIVELQELLEK.[Q] | 3xPhospho [S3; S6; S9] | 3xPhospho [S257; S260; S263] | Up | n.d. | Up |
| 256–279 | [R].LSNGSIDSTDETSQIVELQELLEK.[Q] | 1xPhospho [S/T] | 1xPhospho [S/T] | Up | n.d. | n.d. |
| 256–279 | [R].LSNGSIDSTDETSQIVELQELLEK.[Q] | 2xPhospho [S/T] | 2xPhospho [S/T] | n.d. | n.d. | Up |
| 547–556 | [R].THLDT**S**AELR.[Y] | 1xPhospho [S6] | 1xPhospho [S552] | n.c. | Up | Down |
| 547–570 | [R].THLDTSAELRYSVG**S**LVDSQSDYR.[T] | 1xPhospho [S15] | 1xPhospho [S561] | n.d. | Down | n.c. |
| 666–684 | [R].LIQEEKE**S**TELRAEEIENR.[V] | 1xPhospho [S8] | 1xPhospho [S673] | n.d. | n.d. | Up |

293T cells expressing Liprin-α2 alone or in combination with CASK-WT were lysed, and HA-tagged Liprin was immunoprecipitated using anti-HA magnetic beads. Purified samples were analysed by tryptic digestion, followed by mass spectroscopy. Phosphorylated peptides were identified and quantified in three independent experiments (Exp1-3). "up" and "down" denotes increases and decreases in peptide intensity upon coexpression with CASK, respectively; n.c., no change; n.d., not detected.

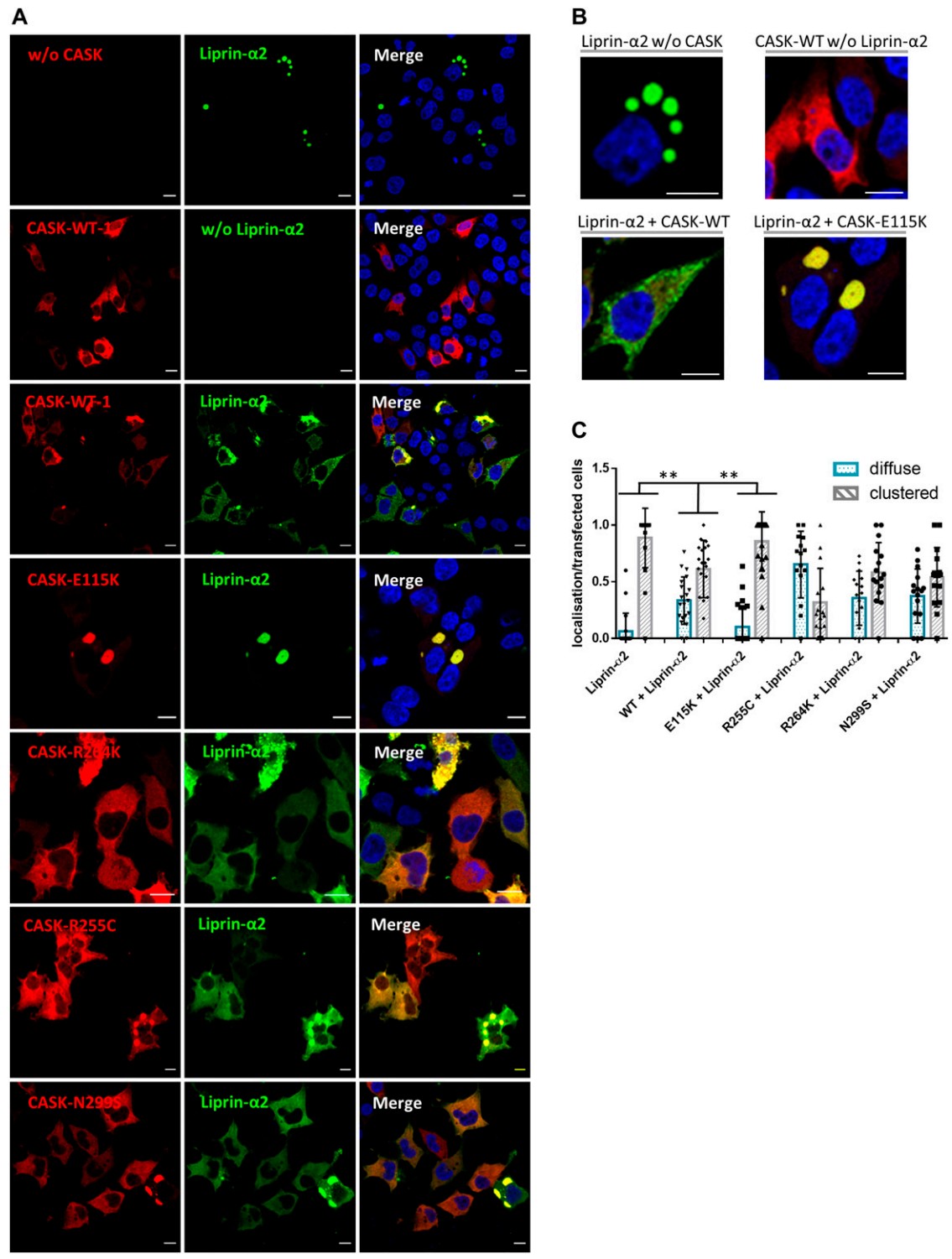

**Figure 6. Coexpression of CASK interferes with formation of spherical condensates by Liprin-α2 in HEK293T cells.**
**(A)** 293T cells expressing GFP-Liprin-α2, mRFP-tagged CASK, or combinations of both proteins were fixed and imaged by confocal microscopy. Images of DAPI-stained nuclei were included in merged pictures. **(B)** Enlargements of cells expressing GFP-Liprin-α2 or mRFP-CASK-WT alone or combinations of GFP-Liprin-α2 with WT or E115K-mutant CASK. **(C)** Quantification of data shown in (A). Five microscopic fields of view were evaluated per independent experiment. All transfected cells were grouped by the criteria "diffuse" or "clustered." Shown is the percentage of cells in each group based on the total number of transfected and analysed cells. For each condition, three independent experiments with a total of more than 90 cells were evaluated. Significant differences from the WT + Liprin-α2 condition were determined by one-way ANOVA followed by Dunnett's multiple comparison test; **$P \leq 0.01$; n = 3.

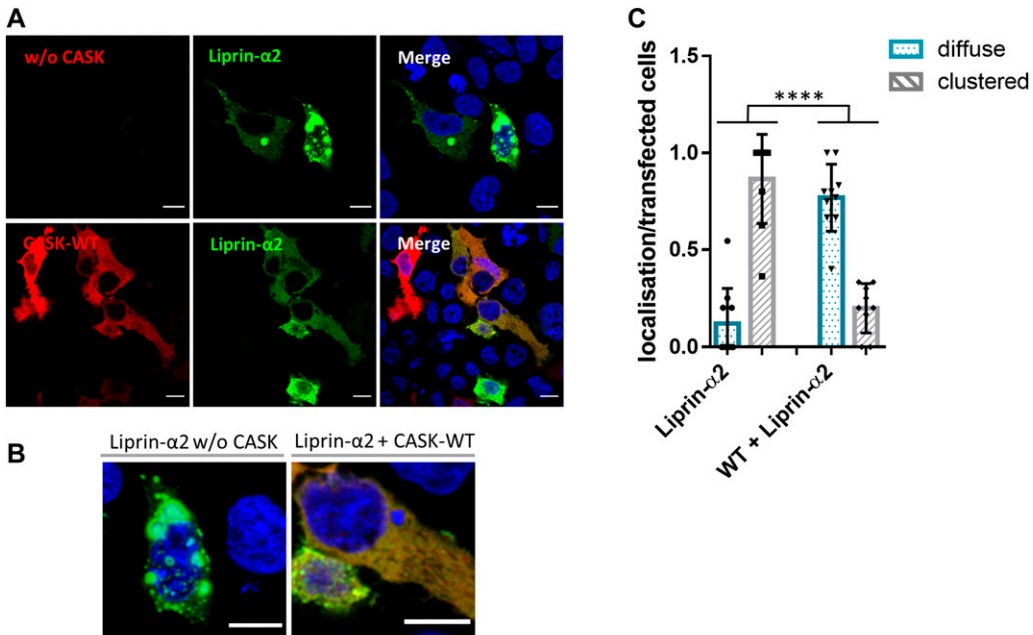

**Figure 7. CASK-WT negatively regulates preformed phase-separated condensates of Liprin-α2.**
**(A)** HEK293T cells were transfected with an expression vector for GFP-tagged Liprin-α2. After 2 d, cells were fixed and processed for confocal microscopy (upper panels). Alternatively, cells were retransfected with mRFP-CASK after 1 d and fixed on the second day (lower panels). **(B)** Enlargement of typical cells shown in (A). **(C)** Quantitative analysis of the data shown in (A); categorization and counting of cells were performed as described in Fig 6.

(McDonald et al, 2020; Emperador-Melero et al, 2021). Clusters of Liprin-α2 in particular adopt properties of phase-separated condensates, as shown by Emperador-Melero et al (2021) and confirmed here by confocal microscopy and FRAP measurements (Fig S3). We observed here that CASK has a regulatory effect on condensate formation. This might depend on direct interaction of the two proteins and/or CASK-Liprin oligomerization as it can be abolished by variants E115K in CASK and W981A in Liprin-α2. Structurally, we do not know how CASK negatively affects condensation of Liprin-α2 into droplet-like clusters. CASK binds to the C-terminal part of Liprin-α2 which has so far not been implicated in condensation. One aspect may be phosphorylation of the N-terminal coiled-coil domain of Liprin-α2 at Ser87, which is reduced upon CASK binding. LLPS of Liprin-α3 is triggered by a phosphorylation event, in this case within the IDR region at Ser760, through the activity of protein kinase C (Emperador-Melero et al, 2021). However, our analysis of Ser87 mutant forms suggest that phosphorylation at this site is not causative for condensation (Fig S4). We currently assume that CASK binding to Liprin-α2 induces a conformational change in Liprin which is incompatible with phase condensation and which alters the accessibility of Ser87 to either kinases or phosphatases which determine the phosphorylation state at this position.

Upon CASK binding, a long α-helical segment (αN-segment in Figs 1 and 5) immediately N-terminal to the SAM domains of Liprin-α performs a significant outward turn (Wei et al, 2011; Xie et al, 2021). As the αN-segment is close in sequence to the IDR, it is conceivable that this movement of αN alters the propensity of the IDR to condensate and induce phase separation.

Data from *C. elegans* suggest that LLPS mediated by Liprin is an essential step in synapse formation; however, the Liprin clusters formed during LLPS are modified at a later stage in synaptogenesis, leading to some form of solidification of the active zone (McDonald et al, 2020). In this respect, our findings that CASK can dissolve preformed phase-separated condensates of Liprin-α2 points to a mechanism where condensate formation is required for early stages of active zone formation, whereas during a later stage, CASK and possibly Neurexin are added to the complex. As depicted by our split-YFP data, this likely occurs in the form of CASK oligomers that would then allow for restructuring of the large Liprin-based condensates. It should be noted that clusters of endogenous Liprin-α variants in neurons are much smaller, in keeping with the size of the presynaptic active zone (Spangler et al, 2011; Zurner et al, 2011).

Importantly, the inability to regulate phase transitions of Liprin-α2 is a functional feature which distinguishes the CASK-E115K variant from the other three investigated variants. E115K caused a PCH phenotype with early lethality, whereas the other three variants did cause a severe neurodevelopmental disorder but without PCH. In further studies, it will be important to delineate how the aberrant regulation of phase-separated condensate formation by Liprin-α proteins contributes to pontocerebellar hypoplasia.

## Materials and Methods

### Patients and genetic analysis

We identified four patients with a *CASK* missense variant from different diagnostic and research cohorts from across India and the United States. Genetic testing was performed by Sanger sequencing of *CASK*, targeted next-generation sequencing gene panel, or exome sequencing. Clinical

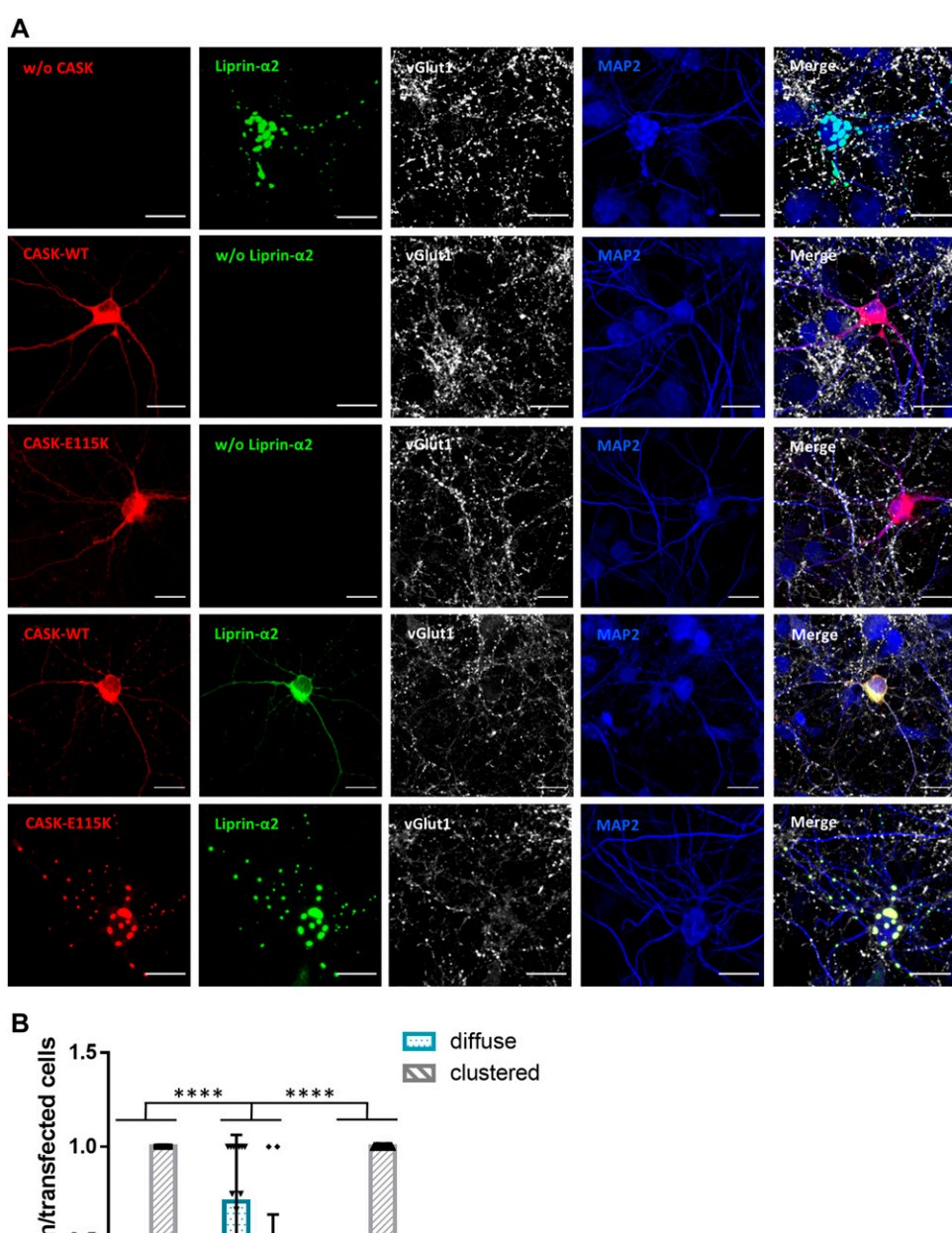

**Figure 8. The presence of CASK interferes with formation of clusters by Liprin-α2 in hippocampal neurons.**
**(A)** Hippocampal neurons transfected with constructs coding for either mRFP-CASK or GFP-Liprin-α2, or both proteins in combination, were fixed and stained for the expressed proteins, as well as vGlut1 as a presynaptic marker and MAP2 as a dendrite marker. **(B)** Quantification of the data shown in (A). Proportion of transfected neurons in which Liprin-α2 was localized diffusely or clustered in the cytoplasm of soma and along the neurites, normalized to the number of transfected cells per image. 15 images per condition were analysed with up to five transfected neurons present, and the mean ± *SD* is shown with each data point representing one analysed picture. Statistical differences were calculated using an ordinary one-way ANOVA with Tukey's test; ****$P \leq 0.0001$.

and molecular findings in patients 1–4 are summarized in Table 1. Informed consent for genetic analysis was obtained from parents/legal guardians, and genetic studies were performed clinically or as approved by the Institutional Review Boards of the respective institution.

Patients for this study were ascertained over a course of 2 yr. Therefore, in some panels of our functional assays, only one or two variants are compared with the respective wild-type condition.

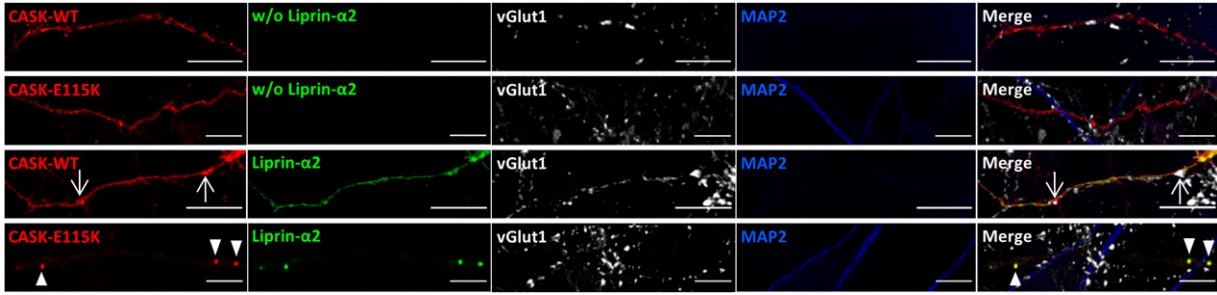

**Figure 9. Liprin-α2 condensate-like axonal clusters are not synaptic.**
Enlargements of axonal segments of transfected neurons shown in Fig 8. Axons were identified by the absence of MAP2 staining. Arrows point to locations of presynaptic sites identified by vGlut1 staining; arrowheads in lower panels point to CASK-Liprin clusters which are devoid of a presynaptic vGlut1 cluster and are therefore considered as non-synaptic.

## Expression constructs

For expression in HEK293T cells, cDNA coding for CASK transcript variant 3 (TV3; (Tibbe et al, 2021)) fused to an N-terminal mRFP-tag in pmRFP-N1 was used. For expression in neurons, cDNAs coding for mRFP-CASK-TV5 fusion proteins were inserted into a vector carrying the human synapsin promoter (Repetto et al, 2018; Tibbe et al, 2021). For the preparation of fusion proteins, the cDNA encoding the CaMK domain was cloned into the pET-SUMO vector coding for a His$_6$-SUMO tag (Thermo Fisher Scientific). An expression vector for GFP-Mint1 was obtained from C Reissner and M. Missler. HA-tagged Neurexin-1β was from P. Scheiffele via Addgene (58267), HA-tagged, and GFP-tagged Liprin-α2 were from C. Hoogenraad. Mutations were introduced using the Quik-Change II site-directed mutagenesis kit (Agilent), using two complementary, mutagenic oligonucleotides. Constructs were verified by Sanger sequencing.

## Antibodies

For Western blotting experiments, all primary antibodies were diluted 1:1,000 in TBS-T with 5% milk powder (MP). The following antibodies were used: α-CASK (Rb, #9497S; Cell Signaling Technologies), α-GFP (Ms, #MMS-118P; Covance), α-Veli 1/2/3 (Rb, #184 002; Synaptic Systems), α-HA (Ms, #H9658; Sigma-Aldrich), and α-Myc (Ms, #M5546, Sigma-Aldrich). HRP-coupled secondary antibodies were used in a dilution of 1:2,500 in TBS-T (Gt-α-Ms or Gt-α-Rb, #BOT-20400 or #BOT-20402; ImmunoReagents). For application in immunocytochemistry of hippocampal neurons, two primary antibodies, both prepared in 1:1,000 dilutions in 2% horse serum

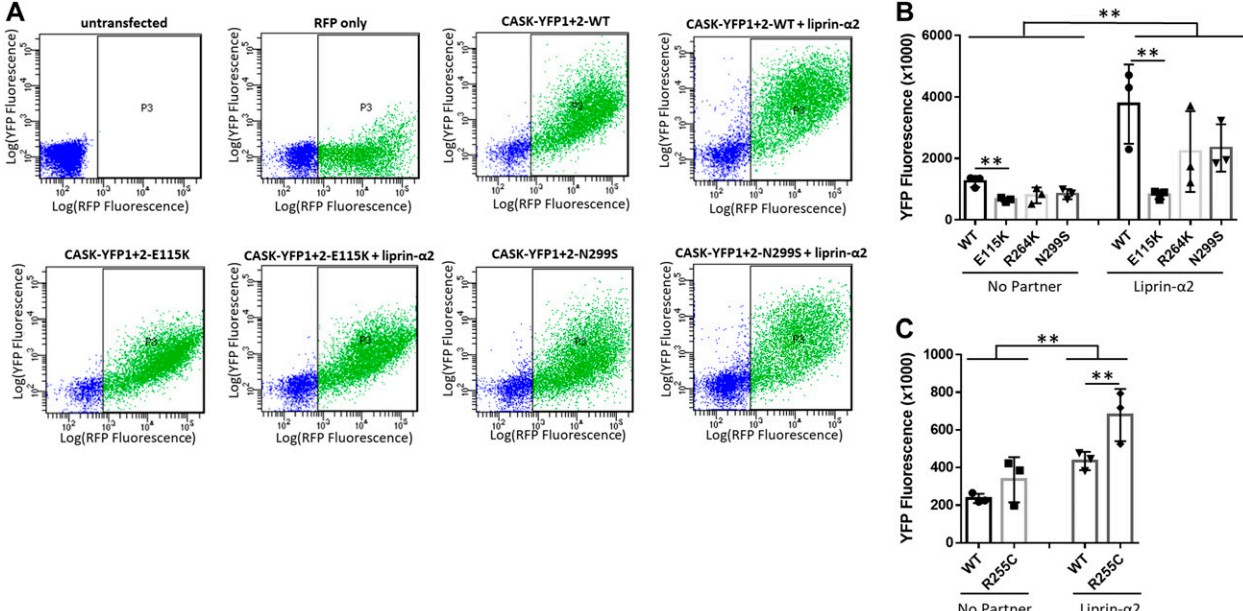

**Figure 10. Liprin-α2 induces formation of CASK oligomers.**
**(A)** HEK293T cells were transfected with plasmids coding for mRFP, CASK (WT or mutant) fused to the N- (YFP1), and the C-terminal (YFP2) halves of YFP and Liprin-α2 as indicated. 2 d after transfection, cells were harvested, resuspended in PBS, and analysed by flow cytometry using filters for mRFP and YFP fluorescence. **(B, C)** Quantification of the data shown in (A). Significance was determined by two-way ANOVA with Sidak's multiple comparison test; **$P \leq 0.01$; ***$P \leq 0.001$; n = 3. Mean ± SD is shown with each data point representing an independent transfection and flow cytometry experiment.

**Figure 11. CASK needs to bind Liprin-α2 to negatively regulate condensate formation.**
**(A)** mRFP-CASK was coexpressed with Liprin-α2-WT or the W981A mutant. After cell lysis and immunoprecipitation of CASK, input (IN) and precipitate (IP) samples were analysed by Western blotting. **(B)** Quantification of the data shown in (A) with mean ± *SD* of three independent transfections depicted by single data points; n = 3. **(C)** Coexpression in HEK293T cells shows that W981A-mutant Liprin-α2 localizes to large intracellular clusters in the absence and in the presence of CASK-WT. **(D)** Quantification of the cell populations shown in (C) based on the total number of transfected cells from 15 images. Shown is the mean ± *SD* with one data point per analysed image. **(E)** Localization of W981A mutant Liprin-α2 and CASK-WT was analysed in transfected primary hippocampal neurons, as before in Fig 7. Here, CASK-WT did not alter the localizaton of the Liprin-α2 mutant. **(F)** Quantification of transfected hippocampal neurons as shown in (E). 10 images per condition were analysed with up to five transfected neurons present and the mean ± *SD* is shown with each data point representing one analysed picture. Statistics was performed with the two-tailed *t* test; ****P ≤ 0.0001; n = 3–6.

(HS) in PBS, were used: $\alpha$-MAP2 (Ck, #ABIN 111 291; Antibodies Online) and $\alpha$-vGlut1 (Rb, #135 303; Synaptic Systems). As secondary antibodies we used Alexa-405 (Gt-$\alpha$-Ck, #ab175675; Abcam) and Alexa-633 (Gt-$\alpha$-Rb, #A-21071; Thermo Fisher Scientific).

## Cell culture, transfection, and coimmunoprecipitation

HEK293T cells (ATCC, CRL-3216) were cultivated on 10-cm dishes in DMEM supplemented with 1× penicillin/streptomycin and 10% foetal bovine serum at 37°C, 5% $CO_2$, and humidified air. Cells were transiently transfected with TurboFect transfection reagent (Thermo Fisher Scientific) according to the manufacturer's instructions. 24 h after transfection, the cells were washed in PBS and lysed in 1 ml of RIPA buffer supplemented with protease inhibitors (0.125 M phenylmethylsulphonyl fluoride, 5 mg/ml leupeptin, 1 mg/ml pepstatin A). Lysates cleared by centrifugation (15 min, 4°C at 20,000$g$) were subjected to immunoprecipitation with 20 $\mu$l of RFP-Trap agarose beads (ChromoTek) for 2 h at 4°C under rotation. The beads were washed five times with RIPA buffer followed by centrifugation (1 min, 4°C, 1,000$g$), and immunoprecipitate (IP) and input samples (IN) were processed for Western blotting. Protein bands were detected by chemoluminescence with a BioRad imaging system in the "auto-mode," avoiding over-saturation while maximizing signal intensity. Band intensities were quantified using Image Lab 6.0 software.

## Bacterial expression and purification of fusion proteins

His$_6$-SUMO–tagged fusion proteins were expressed in BL21 (DE3) cells and purified from bacterial lysates prepared in native lysis buffer (50 mM $NaH_2PO_4$, 500 mM NaCl, pH 8.0) using Ni–NTA agarose (QIAGEN). Proteins were eluted from beads with 250 mM imidazole in lysis buffer and were immediately applied to G-25 columns equilibrated in TNP-ATP–binding buffer (40 mM Tris HCl 100 mM NaCl, 50 mM KCl, pH 7.5), followed by elution in the same buffer. Efficiency of protein purifications was verified by SDS–PAGE, followed by Coomassie staining. Protein concentrations were determined by Bradford assay, using BSA as a standard.

## TNP-ATP binding assay

The binding of 2,4,6-trinitrophenol conjugated ATP (TNP-ATP) to the CaMK domain of CASK was measured by fluorescence spectroscopy. 1 $\mu$M TNP-ATP was added to 100 $\mu$g/ml soluble protein in TNP-ATP binding buffer. For the detection of magnesium sensitivity, 2 mM $Mg^{2+}$ was added. After 15 min incubation, the fluorescence emission spectra from 500 to 600 nm were measured on a Synergy H1 plate reader.

## Cell culture of primary hippocampal neurons

Primary cultures of hippocampal neurons were prepared from *Rattus norvegicus* embryonic day 18 (E18) rats (Wistar Unilever outbred rat, strain: HsdCpb:WU; Envigo) regardless of gender, as described before (Hassani Nia et al, 2020). Neurons were isolated using papain neuron isolation enzyme (#88285; Thermo Fisher Scientific) and cultivated in neurobasal culture media containing B27 and GlutaMAX supplements (#21103049, #17504044 and #A1286001; Thermo Fisher Scientific). Neurons were transfected

using the calcium phosphate method after 7 d in vitro (DIV7), as described before (Hassani Nia et al, 2020). Cells were cultured until DIV14 before fixation and staining.

Animal experiments were approved by, and conducted in accordance with, the guidelines of the Animal Welfare Committee of the University Medical Center under permission number Org1018.

## Immunocytochemistry and confocal microscopy

Transfected HEK293T cells (using 0.05 $\mu$g per plasmid per $cm^2$ of culture dish) were transferred to PLL-coated coverslips in 12-well plates 1 d after transfection and fixed 1 d later. Hippocampal neurons were cultured and transfected on PLL-coated coverslips in 12-well plates. For ICC, cells were washed three times with PBS and fixed in 4% PFA with 4% Sucrose in PBS for 15 min at RT. The cells were washed again three times with cold PBS, followed by permeabilization in 0.1% Triton-X 100 in PBS for 3 min at RT. After washing with PBS, cells were incubated in 10% HS in PBS for 1 h at RT to reduce unspecific antibody binding. Primary antibodies were prepared in 2% HS in PBS, and the cells were incubated with the antibody solution overnight at 4°C in a humidified atmosphere. After washing with PBS, cells were incubated with secondary antibodies diluted in PBS for 1 h at RT. After washing three times with PBS and once with ddH$_2$O, coverslips were mounted with ProLong Diamond Antifade mounting medium (#P36961; Thermo Fisher Scientific). Samples were analysed by confocal microscopy, using a Leica SP8 confocal microscope (provided by UKE Microscopy Imaging Facility; UMIF). For 293T cells, single stacks were recorded; for neurons, z-stacks were analysed. For quantification, phase-separated condensates were counted manually; the evaluating person was blind to the experimental conditions.

## FastAP dephosphorylation assay

HEK293T cells expressing HA-Liprin-$\alpha$2 alone or together with mRFP-CASK-wildtype (WT) were lysed in IP buffer without EDTA. Cleared lysates were treated with or without FastAP Thermosensitive Alkaline Phosphatase (EF0651; Thermo Fisher Scientific), as described in the manufacturer's protocol for protein dephosphorylation. After incubation at 37°C for 1 h under shaking, samples were analysed by immunoblotting.

## Sample preparation for proteome analysis

HA-tagged Liprin-$\alpha$2 was immunoprecipitated from transfected cells using HA-specific magnetic beads. After washing, samples were diluted in 1% wt/vol sodium deoxycholate (SDC) in 100 mM triethyl bicarbonate buffer and boiled at 95°C for 5 min. Disulfide bonds were reduced in the presence of 10 mM DTT at 60°C for 30 min. Cysteine residues were alkylated in presence of 20 mM iodoacetamide at 37°C in the dark for 30 min, and tryptic digestion (sequencing grade; Promega) was performed at a 100:1 protein to enzyme ration at 37°C overnight. Digestion was stopped and SDC precipitated by the addition of 1% vol/vol formic acid (FA). Samples were centrifuged at 16,000$g$ for 5 min, and the supernatant was transferred into a new tube. Samples were dried in a vacuum centrifuge.

### LC–MS/MS in data dependent mode

Samples were resuspended in 0.1% formic acid (FA) and transferred into a full recovery autosampler vial (Waters). Chromatographic separation was achieved on a UPLC system (nanoAcquity; Waters) with a two-buffer system (buffer A: 0.1% FA in water, buffer B: 0.1% FA in ACN). Attached to the UPLC was a C18 trap column (Symmetry C18 Trap Column, 100 Å, 5 $\mu$m, 180 $\mu$m × 20 mm; Waters) for online desalting and sample purification followed by an C18 separation column (BEH130 C18 column, 75 $\mu$m × 25 cm, 130 Å pore size, 1.7 $\mu$m particle size; Waters). Peptides were separation using a 60-min gradient with increasing acetonitrile concentration from 2 to 30%. The eluting peptides were analysed on a quadrupole orbitrap mass spectrometer (QExactive; Thermo Fisher Scientific) in data dependent acquisition.

### Data analysis and processing

Acquired data dependent acquisition LC–MS/MS data were searched against the reviewed human protein database downloaded from Uniprot (release April 2020, 20,365 protein entries, EMBL) using the Sequest algorithm integrated in the Proteome Discoverer software version 2.4 (Thermo Fisher Scientific) in label-free quantification mode with match between runs enabled, performing chromatographic retention re-calibration for precursors with a 5-min retention time tolerance, no scaling, and no normalization for extracted peptide areas was performed. Mass tolerances for precursors were set to 10 ppm and 0.02 D for fragments. Carbamidomethylation was set as a fixed modification for cysteine residues and the oxidation of methionine, phosphorylation of serine and threonine, pyro-glutamate formation at glutamine residues at the peptide N-terminus and acetylation of the protein N-terminus, methionine loss at the protein N-terminus, and the acetylation after methionine loss at the protein N-terminus were allowed as variable modifications. Only peptides with a high confidence (false discovery rate <1% using a decoy database approach) were accepted as identified.

### Split-YFP experiments

HEK293T cells were cotransfected with 3 $\mu$g of CASK-YFP1 and CASK-YFP2 expression vectors, in combination with 1 $\mu$g of pmRFP-C1 plasmid and either 3 $\mu$g of Liprin-$\alpha$2 or empty vector. 2 d later, cells were trypsinized, centrifuged at $500g$ for 5 min, and resuspended in PBS. Flow cytometry was performed at a FACS Canto-II instrument (BD Biosciences). Transfected cells were identified by mRFP fluorescence, and YFP fluorescence was quantified from 10,000 events from viable single cells for each condition. Aliquots of cells were additionally analysed by immunoblotting to determine efficient expression of CASK fusion proteins.

## Supplementary Information

## Acknowledgements

We thank Hans-Hinrich Hönck for excellent technical support, Carsten Reissner and Markus Missler (Münster, Germany), Peter Scheiffele (Basel, Switzerland), and Caspar Hoogenraad (Utrecht, The Netherlands) for plasmids. We thank the UKE microscopy imaging facility (umif) for providing confocal microscopes and assistance with FRAP measurements and data analysis. We thank Jonas Runge for calculating the Kolmogorov-Smirnov test. This work was supported by grants from Deutsche Forschungsgemeinschaft (KU 1240/10-1 to K Kutsche, KR 1321/7-1 to H-J Kreienkamp) and by Jordan's Guardian Angels, the Sunderland Foundation, and the Brotman Baty Institute (to G Mirzaa).

## Author Contributions

D Tibbe: conceptualization, investigation, visualization, and writing—original draft.
P Ferle: investigation and visualization.
C Krisp: validation and investigation.
S Nampoothiri: formal analysis, validation, and investigation.
G Mirzaa: formal analysis, validation, and investigation.
M Assaf: formal analysis, validation, and investigation.
S Parikh: formal analysis, validation, and investigation.
K Kutsche: conceptualization, formal analysis, supervision, funding acquisition, and validation.
H-J Kreienkamp: conceptualization, supervision, funding acquisition, visualization, and writing—review and editing.

## Conflict of Interest Statement

The authors declare that they have no conflict of interest.

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
