## [Reviewer comments · Life Science Alliance]

Life Science Alliance

Regulation of Liprin- α phase separation by CASK is disrupted by a mutation in its CaM kinase domain

Debora Tibbe, Pia Ferle, Christoph Krisp, Sheela Nampoothiri, Ghayda Mirzaa, Melissa Assaf, Sumit Parikh, Kerstin Kutsche, and Hans-Jürgen Kreienkamp

DOI: <https://doi.org/10.26508/lsa.202201512>

Corresponding author(s): Hans-Jürgen Kreienkamp, University Medical Center Hamburg-Eppendorf, Department of Human Genetics

Review Timeline:

Submission Date:	2022-05-03
Editorial Decision:	2022-05-27
Revision Received:	2022-08-22
Editorial Decision:	2022-09-06
Revision Received:	2022-09-08
Accepted:	2022-09-09

Scientific Editor: Novella Guidi

Transaction Report:

May 27, 2022

Re: Life Science Alliance manuscript #LSA-2022-01512-T

Hans-Jürgen Kreienkamp
University Medical Center Hamburg-Eppendorf, Department of Human Genetics
Human Genetics
University Medical Center Hamburg-Eppendorf
Hamburg, Hamburg 20246
Germany

Dear Dr. Kreienkamp,

Thank you for submitting your manuscript entitled "Regulation of Liprin- α phase separation by CASK is disrupted by a mutation in its CaM kinase domain" to Life Science Alliance. The manuscript was assessed by expert reviewers, whose comments are appended to this letter. We invite you to submit a revised manuscript addressing the Reviewer comments.

Thank you for this interesting contribution to Life Science Alliance. We are looking forward to receiving your revised manuscript.

Sincerely,

B. MANUSCRIPT ORGANIZATION AND FORMATTING:

Reviewer #1 (Comments to the Authors (Required)):

Tibbe and co-authors characterize four novel missense mutations in the calcium/calmodulin-dependent kinase domain of CASK, a membrane associated-guanylate kinase. CASK interacts with several scaffolding and transmembrane proteins important in the formation and function of synapses between neurons in the central nervous system. Loss-of-function variants of the X-chromosomal CASK gene lead to microcephaly with pontine and cerebellar hypoplasia and intellectual disability, and the missense mutations characterized in this study replicate the loss-of-function phenotype to varying degrees. These missense mutations are located close to a protein binding site in CASK for the interacting protein family of alpha-liprins and are shown to reduce the binding between CASK and liprin-alpha2. It is known alpha-liprins are capable of liquid-liquid phase separation (LLPS) in neurons as well as in non-neuronal cells. Tibbe and collaborators show that CASK is able to negatively regulate LLPS of alpha-liprin. Interestingly, the mutation with the severest phenotype is unable to interfere with liprin-alpha2 LLPS.

The characterization of the effect of the four CASK missense mutations in the CaMK domain is certainly extensive and carefully performed in the protein interaction and ATP binding parts of the study. The finding that CASK is a negative regulator of LLPS of liprin-alpha2, which the authors chose to emphasize in this manuscript, is interesting and novel. A potential limitation of the work is that the authors are unable to precisely define the negative regulatory mechanism of liprin-alpha phase separation by CASK. CASK binding to liprin-alpha appears to be necessary to prevent LLPS, as data obtained with the liprin-alpha2 W981A mutant suggests. However, CASK binding is not sufficient: All four CASK mutants analyzed in this study reduce CASK/liprin-alpha2 interactions to a similar degree, but only the E115K mutant appears to lack the negative regulatory effect on liprin-alpha2 LLPS. In the discussion, authors implicate effects of CASK on liprin-alpha2 phosphorylation, but the evidence provided is somewhat circumstantial (see point 4 below). While the authors are likely unable to pinpoint the mechanism of inhibition of alpha-liprin LLPS by CASK within the framework of the study, a revision of the manuscript could improve the experimental support of the main conclusions drawn by the authors. Specific suggestions for a revision are:

- 1) Figure 6 claims to show that liprin-alpha2 LLPS in HEK 293T cells is inhibited in the presence of wild-type CASK as well as the CASK missense mutations R255C, R264K and N299S, but not in the presence of the E115K variant. Unfortunately, the analysis methodology chosen complicates the interpretation of the result. The authors classify individual 293T cells expressing liprin-alpha2 into four categories according to the appearance of "large", "small", "mixed small or large" liprin alpha clusters or their absence ("diffuse" staining). No indication is given why cells with apparent liprin-alpha LLPS are further subcategorized or what the criteria of this categorization exactly are. Crucially, no statistical evaluation of the data is given for this important experiment. The authors should re-classify 293T as either exhibiting liprin-alpha LLPS or not, as done in figure 8 for LLPS in neurons, and then analyze their data statistically. If the authors have a rationale for determining the size of the spherical liprin-alpha condensates, they should do so in a separate analysis. Data in figure 7 should be re-analyzed accordingly.
- 2) Figure 8 shows the effect of CASK on LLPS of liprin-alpha2 in neurons, a valuable experiment since neurons express additional liprin-alpha and CASK interacting proteins not found in 293T cells and likely exhibit different cellular signalling, which could lead to altered liprin-alpha LLPS. The experiment has been performed with overexpressed wild-type CASK as well as the E115K variant, but not the three other missense mutations. This is unfortunate because the ability of these mutants to interfere with liprin-alpha LLPS may be quantitatively different in neurons compared to 293T cells. It is also interesting to note that suppression of liprin-alpha LLPS with overexpressed wild-type or mutant CASK occur in the presence of endogenous CASK in neurons. It would be valuable to know if suppression of liprin-alpha LLPS is dependent on the level of wild type CASK overexpression, or the expression ratios of overexpressed CASK and overexpressed liprin alpha2.
- 3) The authors show that liprin-alpha2 binding to CASK increases the ability of the membrane associated-guanylate kinase to oligomerize via its C-terminal PDZ-SH3-GK domain. Intriguingly, the E115K missense mutation, but not the N299S variant, prevents the liprin-binding induced oligomerization. Do the authors have any explanation how a mutation at the N-terminus of CASK affects its ability to oligomerize via its C-terminus? Do they have any data how the other two missense mutations characterized in this manuscript affect the ability of CASK to oligomerize? Could the oligomerization of CASK be mechanistically linked to its negative regulatory role of liprin-alpha LLPS?
- 4) In the discussion, authors strongly suggest that changes in liprin-alpha2 phosphorylation brought about by CASK binding influences LLPS. While I appreciate the parallel drawn to liprin-alpha3 phosphorylation and LLPS in the Emperador-Melero

(2021) study, the data presented here are circumstantial and do not support this notion. The MS data aiming to show a reduction of liprin-alpha2 phosphorylation at S87 in the presence of wild-type CASK (table 2) are not statistically evaluated. The role of S87 phosphorylation in LLPS of liprin-alpha2 is not tested by mutations abolishing or mimicking phosphorylation (S87A, S87D). Moreover, the gel shift of liprin-alpha 2 in SDS-PAGE gels ascribed to phosphorylation occurs in the presence of all three of the missense mutations tested in this experiment, which indicates that any effect of CASK on liprin-alpha2 phosphorylation is abrogated by all missense mutations analyzed in this study, and thus not material to the change in liprin-alpha2 LLPS only seen with the E115K mutation. The discussion of phosphorylation in the context of liprin-alpha2 LLPS should reflect the circumstantial nature of the phosphorylation data.

Reviewer #2 (Comments to the Authors (Required)):

This study by Tibbe and colleagues describes four patients with neurodevelopmental disorders that carry new mutations in the synaptic scaffold protein CASK. The study also shows that all four mutant CASK proteins have reduced ability to co-immunoprecipitate with Liprin- α 2, but mostly not with other known presynaptic binding partners. It is further shown that direct binding of CASK to Liprin- α 2 prevents overexpressed Liprin- α 2 to form spherical droplets in HEK cells, and that at least the most severe CASK mutation (E115K) is incapable of regulating this. Finally, this study shows that the phosphorylation of overexpressed Liprin- α 2 in HEK cells at Ser85 is decreased when CASK is co-expressed.

Together, these experiments lead to the suggestion that the direct binding of CASK to Liprin- α 2 regulates phase condensation, and that the inability to execute such regulation may cause neurodevelopmental disorders. This model is very preliminary, as the evidence provided by the study mostly comes from overexpression experiments. However, in light of the recent evidence that liquid-liquid phase separation of synaptic scaffolds is necessary for the formation of function of synapses (see, for instance PMIDs 33208945, 34031393 or 32015539), I consider that the overall observations of this study deserve to be published. However, there are three major concerns that have to be addressed prior to publication:

1. Liprin- α 2 condensation in HEK cells is not properly measured and quantified. First, the experimental analysis is incomplete, as the authors only describe the formation of spherical condensates by Liprin- α 2, which is just the first indication of phase separation. At the very least, the authors should also measure the exchange of molecules between the condense and dilute phases (the easiest way is to FRAP the condensates) to determine the nature (liquid, gel-like...) of the condensates. While it seems that Liprin- α 2 behave as a gel-like condensate (PMIDs 34031393 & 33761347), it is important to describe the nature of the condensates that are formed in the presence of CASK, especially the mutant forms.

Second, quantifying condensates as diffuse, small, large and mixed seems very unconventional, and prevents the authors to show the real effect of the manipulations. They should provide a more accurate quantification to properly describe how CASK (and mutant CASK proteins) alter Liprin- α 2 condensation. Quantifying the number of condensates per co-transfected cell and the area of the condensates formed would certainly be a much better way.

2. The overexpression experiments in neurons are not particularly insightful. These experiments only show the same as the HEK cell experiments do, which is that overexpressing Liprin- α 2 in a cellular environment results in the formation of puncta. Because transfecting neurons typically results in expression levels orders of magnitude above endogenous levels, this is very far from reflecting the real behavior or distribution of Liprin- α 2 and CASK in neurons, and it is likely that the authors are just reporting overexpression artifacts. As a matter of fact, in their overexpressed neurons, Liprin and CASK are barely punctate, while the literature shows a punctate distribution for endogenous Liprin- α 2 in wild type neurons that also express endogenous levels of CASK (PMIDs 21618222 & 21618221).

The most accurate way to conduct these experiments would be to express proteins of interest (with or without mutations) with a more controlled method (usually lentiviral expression) in a knockout background. If the authors cannot do these experiment, and finally decide to maintain the current overexpression ones, three points need to be amended:

- They have to make it clear that these are transfected neurons at every instance (abstract, introduction, results, figure legends and discussion)
- They have to state at certain point that this may not reflect the endogenous distribution of Liprin- α 2. They also have to compare their overexpression results to the literature where endogenous Liprin has been imaged (at least PMIDs 21618222 & 21618221).
- They have to remove any claim about phase separation or condensation in neurons. The objects that the authors see can only be referred to as "puncta", not even condensates. For all we know, they may be targeted to organelles or to any other compartment.

3. It is very unclear how CASK alters Liprin- α 2 phosphorylation, and how this is relevant for the proposed mechanism. First, it is very unintuitive that a protein without phosphatase activity (CASK) reduces Liprin phosphorylation. It seems likely that the interaction between CASK and Liprin prevents another protein to dephosphorylate Liprin- α 2. Is there any way that the authors can test this?

Second, one wonders whether the phosphorylation stage of Liprin- α 2 determines condensation in HEK cells. This study should address this possibility. The easiest way is to assess whether the phosphodead (S87A) and mimetic (S87E) mutants condensate alone and in the presence of CASK in HEK cells.

Reviewer #1 (Comments to the Authors (Required)):

Tibbe and co-authors characterize four novel missense mutations in the calcium/calmodulin-dependent kinase domain of CASK, a membrane-associated guanylate kinase. CASK interacts with several scaffolding and transmembrane proteins important in the formation and function of synapses between neurons in the central nervous system. Loss-of-function variants of the X-chromosomal CASK gene lead to microcephaly with pontine and cerebellar hypoplasia and intellectual disability, and the missense mutations characterized in this study replicate the loss-of-function phenotype to varying degrees. These missense mutations are located close to a protein binding site in CASK for the interacting protein family of alpha-liprins and are shown to reduce the binding between CASK and liprin-alpha2. It is known alpha-liprins are capable of liquid-liquid phase separation (LLPS) in neurons as well as in non-neuronal cells. Tibbe and collaborators show that CASK is able to negatively regulate LLPS of alpha-liprin. Interestingly, the mutation with the severest phenotype is unable to interfere with liprin-alpha2 LLPS.

The characterization of the effect of the four CASK missense mutations in the CaMK domain is certainly extensive and carefully performed in the protein interaction and ATP binding parts of the study. The finding that CASK is a negative regulator of LLPS of liprin-alpha2, which the authors chose to emphasize in this manuscript, is interesting and novel. A potential limitation of the work is that the authors are unable to precisely define the negative regulatory mechanism of liprin-alpha phase separation by CASK. CASK binding to liprin-alpha appears to be necessary to prevent LLPS, as data obtained with the liprin-alpha2 W981A mutant suggests. However, CASK binding is not sufficient: All four CASK mutants analyzed in this study reduce CASK/liprin-alpha2 interactions to a similar degree, but only the E115K mutant appears to lack the negative regulatory effect on liprin-alpha2 LLPS. In the discussion, authors implicate effects of CASK on liprin-alpha2 phosphorylation, but the evidence provided is somewhat circumstantial (see point 4 below). While the authors are likely unable to pinpoint the mechanism of inhibition of alpha-liprin LLPS by CASK within the framework of the study, a revision of the manuscript could improve the experimental support of the main conclusions drawn by the authors. Specific suggestions for a revision are:

1) Figure 6 claims to show that liprin-alpha2 LLPS in HEK 293T cells is inhibited in the presence of wild-type CASK as well as the CASK missense mutations R255C, R264K and N299S, but not in the presence of the E115K variant. Unfortunately, the analysis methodology chosen complicates the interpretation of the result. The authors classify individual 293T cells expressing liprin-alpha2 into four categories according to the appearance of "large", "small", "mixed small or large" liprin alpha clusters or their absence ("diffuse" staining). No indication is given why cells with apparent liprin-alpha LLPS are further subcategorized or what the criteria of this categorization exactly are. Crucially, no statistical evaluation of the data is given for this important experiment. The authors should re-classify 293T as either exhibiting liprin-alpha LLPS or not, as done in figure 8 for LLPS in neurons, and then analyze their data statistically. If the authors have a rationale for determining the size of the spherical liprin-alpha condensates, they should do so in a separate analysis. Data in figure 7 should be re-analyzed accordingly.

Response: We thank this reviewer for the positive and encouraging comments. We have followed the advice of this reviewer and have eliminated the different categories in Fig. 6; the statistical analysis clearly shows significant differences between the "Liprin + WT CASK" situation, and the "Liprin, no CASK" and "Liprin + CASK-E115K" conditions. Here the E115K variant behaves similar to "no CASK", fully supporting our initial conclusions. No significant differences are found for the other three

mutants in the analysis of diffuse vs. clustered population of cells. A more detailed analysis was also included as a new supplementary Figure S2, based on cumulative plots for the size of the clusters, and the number of the clusters per cells, where the E115K, R264K and N299S variants showed significant differences in terms of cluster size compared to the CASK-WT.

2) Figure 8 shows the effect of CASK on LLPS of liprin-alpha2 in neurons, a valuable experiment since neurons express additional liprin-alpha and CASK interacting proteins not found in 293T cells and likely exhibit different cellular signalling, which could lead to altered liprin-alpha LLPS. The experiment has been performed with overexpressed wild-type CASK as well as the E115K variant, but not the three other missense mutations. This is unfortunate because the ability of these mutants to interfere with liprin-alpha LLPS may be quantitatively different in neurons compared to 293T cells. It is also interesting to note that suppression of liprin-alpha LLPS with overexpressed wild-type or mutant CASK occur in the presence of endogenous CASK in neurons. It would be valuable to know if suppression of liprin-alpha LLPS is dependent on the level of wild type CASK overexpression, or the expression ratios of overexpressed CASK and overexpressed liprin alpha2.

Response: For the revised version, we have performed extensive experiments in neurons for the other three mutants. The data show indeed very significant effects of the R255C and the N299S mutant, whereas the R264K variant does not differ from CASK-WT. These data have now been included as the supplemental Figure S5.

In addition, we have varied the level of the Liprin-alpha, as well as the level of CASK cDNA in neuronal transfection experiments to determine how different expression levels of both proteins affect the ability to form cellular clusters. These experiments show that at lower levels, Liprin exhibits a reduced ability to aggregate in LLPS-like clusters. On the other hand, CASK can reduce cluster formation even at lower expression levels. We must note here also that, at lower plasmid concentrations, the number of transfected cells is reduced, making it rather difficult to obtain a sufficient number of cells for analysis. These data have now been included as the supplemental Figure S6.

3) The authors show that liprin-alpha2 binding to CASK increases the ability of the membrane associated-guanylate kinase to oligomerize via its C-terminal PDZ-SH3-GK domain. Intriguingly, the E115K missense mutation, but not the N299S variant, prevents the liprin-binding induced oligomerization. Do the authors have any explanation how a mutation at the N-terminus of CASK affects its ability to oligomerize via its C-terminus? Do they have any data how the other two missense mutations characterized in this manuscript affect the ability of CASK to oligomerize? Could the oligomerization of CASK be mechanistically linked to its negative regulatory role of liprin-alpha LLPS?

Response: In the revised version, we have included new experiments for the R255C and R264K variants, as requested by this reviewer. The data show that both variants are more similar to WT and induce an increase in oligomerization, similar to WT.

We have also added a section in the discussion on the presumed interplay between oligomerization and LLPS-like condensate formation. As already indicated in the initial version, we assume that there are two conformational states for Liprin- α 2: the Liprin-CASK-WT complex which is characterized by diffusely localized oligomers; and Liprin- α 2 alone which forms LLPS-like condensates

4) In the discussion, authors strongly suggest that changes in liprin-alpha2 phosphorylation brought about by CASK binding influences LLPS. While I appreciate the parallel drawn to liprin-alpha3 phosphorylation and LLPS in the Emperador-Melero (2021) study, the data presented here are

circumstantial and do not support this notion. The MS data aiming to show a reduction of liprin-alpha2 phosphorylation at S87 in the presence of wild-type CASK (table 2) are not statistically evaluated. The role of S87 phosphorylation in LLPS of liprin-alpha2 is not tested by mutations abolishing or mimicking phosphorylation (S87A, S87D). Moreover, the gel shift of liprin-alpha 2 in SDS-PAGE gels ascribed to phosphorylation occurs in the presence of all three of the missense mutations tested in this experiment, which indicates that any effect of CASK on liprin-alpha2 phosphorylation is abrogated by all missense mutations analyzed in this study, and thus not material to the change in liprin-alpha2 LLPS only seen with the E115K mutation. The discussion of phosphorylation in the context of liprin-alpha2 LLPS should reflect the circumstantial nature of the phosphorylation data.

Response: We agree that the evidence for a functional role of phosphorylation was circumstantial in our initial version of the manuscript. To improve this, and to analyse any potential causality for the cluster formation process, we performed several additional experiments for the revised version.

a. we performed a quantitative analysis of the gel shift phenomenon, now shown in the new supplemental Figure S1. Here we split the Liprin-specific band in a "tail" and a "front" half, and compared the relative abundance of signal in the "tail" section. These data show that tailing in the "no CASK" and CASK-E115K situations is significantly different from the wild type situation, validating our previous conclusions. For the other mutants, intermediate results were obtained which did not significantly differ from either the "no CASK" or the CASK-WT situation.

b. as suggested, we have introduced phospho-mimic and phospho-negative mutations at this position in Liprin. Both variants do not have any significant effect on cluster formation. These data are shown in the new supplemental Figure S4. We have altered our discussion of this phosphorylation event accordingly, stating that this phosphorylation is altered in the presence of CASK, but lacks causality for the clustering of Liprin in LLPS-like structures.

Reviewer #2 (Comments to the Authors (Required)):

This study by Tibbe and colleagues describes four patients with neurodevelopmental disorders that carry new mutations in the synaptic scaffold protein CASK. The study also shows that all four mutant CASK proteins have reduced ability to co-immunoprecipitate with Liprin- α 2, but mostly not with other known presynaptic binding partners. It is further shown that direct binding of CASK to Liprin- α 2 prevents overexpressed Liprin- α 2 to form spherical droplets in HEK cells, and that at least the most severe CASK mutation (E115K) is incapable of regulating this. Finally, this study shows that the phosphorylation of overexpressed Liprin- α 2 in HEK cells at Ser85 is decreased when CASK is co-expressed.

Together, these experiments lead to the suggestion that the direct binding of CASK to Liprin- α 2 regulates phase condensation, and that the inability to execute such regulation may cause neurodevelopmental disorders. This model is very preliminary, as the evidence provided by the study mostly comes from overexpression experiments. However, in light of the recent evidence that liquid-liquid phase separation of synaptic scaffolds is necessary for the formation of function of synapses (see, for instance PMIDs 33208945, 34031393 or 32015539), I consider that the overall observations of this study deserve to be published. However, there are three major concerns that have to be addressed prior to publication:

1. Liprin- α 2 condensation in HEK cells is not properly measured and quantified. First, the

experimental analysis is incomplete, as the authors only describe the formation of spherical condensates by Liprin- α 2, which is just the first indication of phase separation. At the very least, the authors should also measure the exchange of molecules between the condense and dilute phases (the easiest way is to FRAP the condensates) to determine the nature (liquid, gel-like...) of the condensates. While it seems that Liprin- α 2 behave as a gel-like condensate (PMIDs 34031393 & 33761347), it is important to describe the nature of the condensates that are formed in the presence of CASK, especially the mutant forms.

Response: We appreciate the constructive and helpful comments by reviewer 2. We followed the advice of this reviewer and performed FRAP experiments, similar to those described by Emperador-Melero et al (PMID 34031393). Our data indicate that Liprin- α 2 behaves like a droplet-like condensate under all conditions tested, in agreement with the data by Emperador-Melero et al.

Second, quantifying condensates as diffuse, small, large and mixed seems very unconventional, and prevents the authors to show the real effect of the manipulations. They should provide a more accurate quantification to properly describe how CASK (and mutant CASK proteins) alter Liprin- α 2 condensation. Quantifying the number of condensates per co-transfected cell and the area of the condensates formed would certainly be a much better way.

Response: As both reviewers criticized our way of quantifying condensates, we have eliminated the different categories in Fig. 6 (see also response to reviewer 1). In addition we have quantified the number of condensates, as well as the size of condensates under each condition (new supplemental Figure S2).

2. The overexpression experiments in neurons are not particularly insightful. These experiments only show the same as the HEK cell experiments do, which is that overexpressing Liprin- α 2 in a cellular environment results in the formation of puncta. Because transfecting neurons typically results in expression levels orders of magnitude above endogenous levels, this is very far from reflecting the real behavior or distribution of Liprin- α 2 and CASK in neurons, and it is likely that the authors are just reporting overexpression artifacts. As a matter of fact, in their overexpressed neurons, Liprin and CASK are barely punctate, while the literature shows a punctate distribution for endogenous Liprin- α 2 in wild type neurons that also express endogenous levels of CASK (PMIDs 21618222 & 21618221).

The most accurate way to conduct these experiments would be to express proteins of interest (with or without mutations) with a more controlled method (usually lentiviral expression) in a knockout background. If the authors cannot do these experiment, and finally decide to maintain the current overexpression ones, three points need to be amended:

- They have to make it clear that these are transfected neurons at every instance (abstract, introduction, results, figure legends and discussion)

Response: Lentiviral expression of two proteins (CASK and Liprin) in a ko background is clearly beyond our experimental means. Therefore we have added the term “transfected” to all neuron experiments.

- They have to state at certain point that this may not reflect the endogenous distribution of Liprin- α 2. They also have to compare their overexpression results to the literature where endogenous Liprin has been imaged (at least PMIDs 21618222 & 21618221).

Response: We have include these references, and mentioned their results now in our discussion section.

- They have to remove any claim about phase separation or condensation in neurons. The objects that the authors see can only be referred to as "puncta", not even condensates. For all we know, they may be targeted to organelles or to any other compartment.

Response: We followed this suggestion; however, instead of puncta we chose the term "cluster", because we felt that this more accurately reflected our observations.

3. It is very unclear how CASK alters Liprin- α 2 phosphorylation, and how this is relevant for the proposed mechanism. First, it is very unintuitive that a protein without phosphatase activity (CASK) reduces Liprin phosphorylation. It seems likely that the interaction between CASK and Liprin prevents another protein to dephosphorylate Liprin- α 2. Is there any way that the authors can test this?

Response: More proteomic data will be needed to identify whether any phosphatase is in complex with CASK. Currently, we find this to be a very worthwhile, but also experimentally very difficult endeavour, which may be outside the scope of this manuscript. We have addressed this issue in the discussion; importantly, we think that interaction with CASK, and the concurrent shift from LLPS-like condensates to a diffuse distribution, will change the accessibility of Liprin- α 2 for phosphatases. This might be responsible for the altered phosphorylation status.

Second, one wonders whether the phosphorylation stage of Liprin- α 2 determines condensation in HEK cells. This study should address this possibility. The easiest way is to assess whether the phosphodead (S87A) and mimetic (S87E) mutants condensate alone and in the presence of CASK in HEK cells.

Response: This was also suggested by reviewer 1, and we have addressed this experimentally in the supplemental Figure S4. Our data clearly show that phosphorylation at Ser87 does not determine condensation of Liprin- α 2. In fact these observations fit quite well to the argument that the condensation of Liprin- α 2 might be the reason for altered phosphorylation, and not vice versa.

September 6, 2022

RE: Life Science Alliance Manuscript #LSA-2022-01512-TR

Prof. Hans-Jürgen Kreienkamp
University Medical Center Hamburg-Eppendorf, Department of Human Genetics
Human Genetics
University Medical Center Hamburg-Eppendorf
Hamburg, Hamburg 20246
Germany

Dear Dr. Kreienkamp,

Thank you for submitting your revised manuscript entitled "Regulation of Liprin- α phase separation by CASK is disrupted by a mutation in its CaM kinase domain". We would be happy to publish your paper in Life Science Alliance pending final revisions necessary to meet our formatting guidelines.

- please address the final Reviewer 2' points
- please upload your supplementary figures as single files and add your supplementary figure legends to the main manuscript text
- please add the author contributions to the main manuscript text
- we encourage you to introduce your panels in your figure legends in alphabetical order
- please add a callout for Figure 6C to your main manuscript text

Figure Check:

- Figure 8 legend: there is panel C explained in the legend but Panel C is not in the figure, please correct
- Figure 11 legend: there are panels G,H explained in the legend but these are not in the figure, please correct

A. FINAL FILES:

B. MANUSCRIPT ORGANIZATION AND FORMATTING:

Sincerely,

Reviewer #1 (Comments to the Authors (Required)):

The authors have addressed the concerns outlined in the initial reviews and considerably improved the manuscript. I recommend the publication of this study in LSA.

Reviewer #2 (Comments to the Authors (Required)):

In this revised version, Tibbe and colleagues address most of the issues raised on the initial manuscript. In my opinion, no more experiments are needed. However, there are still few inaccuracies. This can be fixed simply by re-writing the appropriate parts. This includes the following:

- The terminology used to describe phase separation is sometimes inaccurate. First, the manuscript does not properly distinguish between liquid and gel-like condensates. This is problematic because, while certain Liprin- α protein (including vertebrate Liprin- α 3 and the invertebrate *syd2*) have been shown to have liquid properties, both literature and the experiments shown by the authors suggest that Liprin- α 2 forms gel-like condensates. For this reason, I believe they should not use the term "liquid" or hint that Liprin- α 2 formed via "liquid-liquid" phase separation. I suggest they simply refer to the Liprin- α 2 spherical droplets as phase condensates or phase separated condensates. Second, it seems that the authors refer to Liprin condensates as "aggregates" occasionally. I would strongly advise to avoid this, as aggregates are often understood as clumps or non-functional proteins.
- For reproducibility purposes, can the authors provide a detailed explanation about how the analysis of localization/transfected cells was done? Specifically, report the amount of transfected DNA per surface unit, the time range after transfection when images were acquired, minimum size of condensates included, single vs z-stack, and mention whether the analysis was automated or done manually, and blind or unblind.
- The relationship between CASK binding to Liprin- α 2 and condensate formation remains slightly unclear. On the one hand, the authors show that phase condensates formed by the W981A Liprin- α 2 mutant (which does not immunoprecipitate with CASK, as opposed to with wild type Liprin) are not regulated by CASK co-transfection. On the other hand, the W981A Liprin- α 2 mutant can recruit CASK to condensates and, furthermore, while the four described patient mutations (E115K, R255, R264 and Y268) strongly reduce binding to Liprin- α 2, only one of them significantly affects condensate formation in transfected cells. Together, this suggest that some form of binding remains between these two proteins and that regulation of phase condensation may be

executed via a more subtle mechanism. I believe that the authors should not make any strong claims about condensate formation being reversed by CASK interaction with Liprin- α 2 (especially in the abstract and introduction). Instead, the likelihood of this could be addressed in the discussion section.

thanks for the positive response to the revised version of our manuscript. We are now uploading a hopefully final version of this manuscript. We have dealt with your comments and requests in the following way:

1. we have addressed the final points from reviewer 2 (see below)
2. supplementary files are uploaded as single files; the legends are now in the main manuscript text.
3. Author contributions have been added before the acknowledgements.
4. we have made sure that all figure panels are listed in alphabetical order in the respective legends.
5. Figure 6C is now mentioned in the main text.
6. For Figures 8 and 11, the legend now fits to the Figure.

Regarding reviewer 2:

The terminology used to describe phase separation is sometimes inaccurate. First, the manuscript does not properly distinguish between liquid and gel-like condensates. This is problematic because, while certain Liprin- α protein (including vertebrate Liprin- α 3 and the invertebrate syd2) have been shown to have liquid properties, both literature and the experiments shown by the authors suggest that Liprin- α 2 forms gel-like condensates. For this reason, I believe they should not use the term "liquid" or hint that Liprin- α 2 formed via "liquid-liquid" phase separation. I suggest they simply refer to the Liprin- α 2 spherical droplets as phase condensates or phase separated condensates. Second, it seems that the authors refer to Liprin condensates as "aggregates" occasionally. I would strongly advise to avoid this, as aggregates are often understood as clumps or non-functional proteins.

Response: we have eliminated "aggregates", and have chosen the term phase separated condensates throughout

For reproducibility purposes, can the authors provide a detailed explanation about how the analysis of localization/transfected cells was done? Specifically, report the amount of transfected DNA per surface unit, the time range after transfection when images were acquired, minimum size of condensates included, single vs z-stack, and mention whether the analysis was automated or done manually, and blind or unblind.

Response: all these details have now been added in the relevant Materials and Methods section.

The relationship between CASK binding to Liprin- α 2 and condensate formation remains slightly unclear. On the one hand, the authors show that phase condensates formed by the W981A Liprin- α 2 mutant (which does not immunoprecipitate with CASK, as opposed to with wild type Liprin) are not regulated by CASK co-transfection. On the other hand, the W981A Liprin- α 2 mutant can recruit CASK to condensates and, furthermore, while the four described patient mutations (E115K, R255, R264 and Y268) strongly reduce binding to Liprin- α 2, only one of them significantly affects condensate formation in transfected cells. Together, this suggests that some form of binding remains between these two proteins and that regulation of phase condensation may be executed via a more subtle mechanism. I believe that the authors should not make any strong claims about condensate formation being reversed by CASK interaction with Liprin- α 2 (especially in the abstract and introduction). Instead, the likelihood of this could be addressed in the discussion section.

Response: We have removed these strong claims in Abstract and Introduction, and also softened our wording at one point in the Discussion.

September 9, 2022

RE: Life Science Alliance Manuscript #LSA-2022-01512-TRR

Prof. Hans-Jürgen Kreienkamp
University Medical Center Hamburg-Eppendorf, Department of Human Genetics
Human Genetics
University Medical Center Hamburg-Eppendorf
Hamburg, Hamburg 20246
Germany

Dear Dr. Kreienkamp,

Thank you for submitting your Research Article entitled "Regulation of Liprin- α phase separation by CASK is disrupted by a mutation in its CaM kinase domain". It is a pleasure to let you know that your manuscript is now accepted for publication in Life Science Alliance. Congratulations on this interesting work.

DISTRIBUTION OF MATERIALS:

Again, congratulations on a very nice paper. I hope you found the review process to be constructive and are pleased with how the manuscript was handled editorially. We look forward to future exciting submissions from your lab.

Sincerely,
